# Private and Fair Machine Learning: Revisiting the Disparate Impact of Differentially Private SGD

**Lea Demelius**                                          *ldemelius@know-center.at*
*Know Center Research GmbH, Graz, Austria*
*Graz University of Technology, Graz, Austria*

**Dominik Kowald**                                        *dkowald@know-center.at*
*Know Center Research GmbH, Graz, Austria*
*Graz University of Technology, Graz, Austria*

**Simone Kopeinik**                                       *skopeinik@know-center.at*
*Know Center Research GmbH, Graz, Austria*

**Roman Kern**                                            *rkern@know-center.at*
*Know Center Research GmbH*
*Graz University of Technology, Graz, Austria*

**Andreas Trügler**                                       *andreas.truegler@uni-graz.at*
*Know Center Research GmbH, Graz, Austria*
*University of Graz, Graz, Austria*

**Reviewed on OpenReview:** *https://openreview.net/forum?id=o8zrx0bfTp*

## Abstract

Differential privacy (DP) is a prominent method for protecting information about individuals during data analysis. Training neural networks with differentially private stochastic gradient descent (DPSGD) influences the model's learning dynamics and, consequently, its output. This can affect the model's performance and fairness. While the majority of studies on the topic report a negative impact on fairness, it has recently been suggested that fairness levels comparable to non-private models can be achieved by optimizing hyperparameters for performance directly on differentially private models (rather than re-using hyperparameters from non-private models, as is common practice). In this work, we analyze the generalizability of this claim by 1) comparing the disparate impact of DPSGD on different performance metrics, and 2) analyzing it over a wide range of hyperparameter settings. We highlight that a disparate impact on one metric does not necessarily imply a disparate impact on another. Most importantly, we show that while optimizing hyperparameters directly on differentially private models does not mitigate the disparate impact of DPSGD reliably, it can still lead to improved utility-fairness trade-offs compared to re-using hyperparameters from non-private models. We stress, however, that any form of hyperparameter tuning entails additional privacy leakage, calling for careful considerations of how to balance privacy, utility and fairness. Finally, we extend our analyses to DPSGD-Global-Adapt, a variant of DPSGD designed to mitigate the disparate impact on accuracy, and conclude that this alternative may not be a robust solution with respect to hyperparameter choice.

## 1 Introduction

The widespread presence of artificial intelligence (AI) and its concurrent influence on society has brought increasing attention to the necessity to develop trustworthy AI systems (Kaur et al., 2022; Kowald et al.,

2024). Among the key requirements are privacy-preservation and fairness (Smuha, 2019; Kaur et al., 2021; Li et al., 2023). While both of these aspects are whole research fields on their own, increased effort is put towards understanding their interrelations, synergies and trade-offs.

One prominent method for preserving the privacy of training data in ML models is differential privacy (DP) (Dwork, 2006; Dwork & Roth, 2014) - a mathematical notion of privacy that makes it possible to learn patterns from the data while protecting information on individuals. For neural networks, the most commonly used method to implement DP is differentially private stochastic gradient descent (DPSGD) (Abadi et al., 2016). It is well known that DPSGD, and DP in general, negatively influence the utility of the computation. While this privacy-utility trade-off is inherent to DP, various techniques - from stringent privacy accounting (Abadi et al., 2016) to machine learning best practices (including hyperparameter tuning) (Papernot et al., 2021) - have been proposed to keep the utility loss acceptably small even for meaningful levels of privacy.

However, Bagdasaryan et al. (2019) have shown that DPSGD has a *disparate impact* on accuracy: the accuracy decreases disproportionately across groups (e.g., women are affected more than men). Follow-up studies (Farrand et al., 2020; Xu et al., 2021; Tran et al., 2021) confirmed this observation, showing that the effect even occurs with balanced group representations and loose privacy guarantees. In contrast, de Oliveira et al. (2023) suggest that DPSGD does not necessarily have a negative impact on fairness, as long as the DP model's hyperparameters are optimized for performance. They infer that the disparate impact of DPSGD primarily occurs when hyperparameter settings that perform well for non-private models are re-used for DP models without further tuning. While this finding would greatly benefit practitioners aiming to train private and fair neural networks, our experiments cannot confirm this claim. Firstly, in contrast to previous work, de Oliveira et al. (2023) uses the area under the ROC curve (AUC-ROC) to measure the models' performance and performance equality rather than accuracy. The question arises, therefore, if accuracy and AUC inequalities always coincide, and if their finding translates to the disparate impact on accuracy. Secondly, the question remains if tuning the clipping norm would suffice in mitigating DPSGD's negative effect on fairness. In this paper, we thouroughly analyze the impact of metric and hyperparameter choice on the disparate impact of DPSGD by answering the following research questions:

RQ1) How does the disparate impact of DPSGD manifest across metrics beyond accuracy such as area under the curve, precision, acceptance rate and error rate, and do these disparities co-occur?

RQ2) How dependent are these disparities on the choice of hyperparameters, and how effective and reliable is hyperparameter tuning in developing private models with similar (or even better) performance and fairness than non-private models?

RQ3) How does hyperparameter choice affect DPSGD-Global-Adapt (Esipova et al., 2022), a variant of DPSGD specifically designed to mitigate the disparate impact of DP?

To answer these three questions, we first give an overview of the relevant literature (Section 2), paying particular attention to the usage of different performance and fairness metrics. Next, we describe the methodologies employed for our analysis (Section 3) and empirically study the impact of DPSGD on various fairness metrics on five datasets, including both tabular and image data (Section 4). We then investigate the influence of the choice of hyperparameters on the impact of DPSGD on performance and fairness by analyzing our results over a wide range of hyperparameter settings, including those optimized for performance (Section 5). Next, we extend our analyses to DPSGD-Global-Adapt, a variant of DPSGD designed to mitigate the disparate impact on accuracy (Section 6). Finally, we address the limitations of our findings (Section 7), and present our conclusions along with directions for future research (Section 8).

## 2 Preliminaries and Related Work

### 2.1 Differentially Private Stochastic Gradient Descent

Differential Privacy (DP) (Dwork, 2006; Dwork & Roth, 2014) is a mathematical definition of privacy for data analysis with the goal of protecting information about individuals while allowing general learnings from their

combined data. A probabilistic algorithm $\mathcal{M} : \mathcal{D} \rightarrow \mathcal{R}$ with domain $\mathcal{D}$ and range $\mathcal{R}$ is $(\epsilon, \delta)$-differentially private if for any two datasets $x, y \in \mathcal{D}$ differing on at most one data point, and any subset of outputs $\mathcal{S} \subseteq \mathcal{R}$, it holds that

$$Pr[\mathcal{M}(x) \in \mathcal{S}] \leq e^\epsilon Pr[\mathcal{M}(y) \in \mathcal{S}] + \delta \tag{1}$$

where $\epsilon$ is the privacy loss (also referred to as the privacy budget) and $\delta$ is the failure probability. The lower $\epsilon$, the stronger the privacy guarantee. $\delta$ is typically set to less than the inverse of the dataset size (Ponomareva et al., 2023).

While DP was initially developed for statistical analysis, it was subsequently adopted for training machine learning models upon realizing that information about individual training data points can be inferred from the model's output (Fredrikson et al., 2015; Shokri et al., 2017; Carlini et al., 2019). The most prominent method for training DP (deep) neural networks is Differentially Private Stochastic Gradient Descent (DPSGD). This technique, a private variant of classical Stochastic Gradient Descent (SGD), ensured DP by clipping the per-example gradients to bound the maximum influence one data point can have on the model and then adding Gaussian noise to the gradients. These changes affect the model's learning dynamics and, consequently, properties such as its performance and fairness.

The privacy-utility trade-off is inherent to DP (due to the addition of noise), but various techniques can help balance it. For DPSGD, the most important aspects are stringent privacy accounting methods (Abadi et al., 2016) and hyperparameter optimization, including architecture choices such as the activation function (Papernot et al., 2021).

## 2.2 Measuring Fairness

Fairness is among the key requirements of trustworthy AI (e.g., (Smuha, 2019; Kaur et al., 2022)). In order to ensure this quality and promote fair treatment of all affected population groups, it requires the measurement of fairness. This guides the development of equitable systems that do not disproportionately affect any group, prioritizing those who have historically faced discrimination based on personal attributes such as race, gender, age, or religion. However, the understanding of fairness in a societal context (with and without AI) encompasses a variety of interpretations, with no consensus on when to apply which (Saxena, 2019). The situation is further complicated by the fact that some notions of fairness are mutually exclusive (Verma & Rubin, 2018). On a fundamental level, fairness and fairness metrics can be categorized as individual and group fairness that relate to the legal concepts of disparate treatment and disparate impact, respectively (Barocas & Selbst, 2016). Hereinafter, we will focus on the concept of group fairness. The following metrics are commonly used to measure group fairness in machine learning:[1]

- Performance equality: The model exhibits equal performance (e.g., *accuracy, AUC-ROC*, or *AUC-PR*) for both groups.
- Statistical/Demographic parity: The model predicts a positive outcome with equal probability for both groups, i.e., both groups have the same *acceptance rate.*
- Predictive parity: The model shows an equal positive predictive value (= *precision*) for both groups, i.e., the same percentage of positive predictions are correct.
- Predictive equality: The model's *false positive error rate*, i.e., the percentage of negative examples that are predicted positive, is equal for both groups.
- Equal opportunity: The model's *false negative error rate*, i.e., the percentage of positive examples are predicted negative, is equal for both groups.
- Equalized odds: This metric combines predictive equality and equal opportunity, i.e., both *false positive* and *false negative error rate* are equal for both groups.

For further details, we point our readers to Verma & Rubin (2018), which provides an illustrative demonstration of these (and other) fairness definitions.

---

[1]We explain the metrics based on binary groups here, given that our work is limited to such; however, metrics can be extended to multiple groups.

## 2.3 Fairness under DPSGD

The disparate impact of DPSGD on model accuracy was first shown by Bagdasaryan et al. (2019). They demonstrated the effect for different use-cases, e.g., gender and age classification of face images and sentiment analysis of tweets. They identified group imbalance as a main driver for the effect, showing that underrepresented groups produce larger gradients, which in turn leads to being impacted more by the gradient clipping. They also investigated the individual effects of various hyperparameters - including clipping norm, noise level, batch size, number of training epochs and the size of underrepresented group. In a follow-up study, Farrand et al. (2020) demonstrated that also small group imbalances in the training data can exhibit a disparate impact of DPSGD on accuracy, and Xu et al. (2021) showed that even overrepresented groups can disproportionately be affected if they have larger average gradient norms (e.g., resulting from a higher complexity of the data distribution). The Farrand et al. (2020) also studied a wider range of privacy budgets and concluded that even loose privacy guarantees can lead to a disparate impact on accuracy. Tighter privacy guarantees can increase fairness as the model becomes more random, i.e., at the cost of overall accuracy. They also reported a disparate impact of DPSGD on equal opportunity and (in some settings) on demographic parity for the CelebA dataset (Liu et al., 2015). The Hansen et al. (2022) confirmed a disparate impact of DPSGD on accuracy for CelebA, and showed a disparate impact on F1 scores for two text datasets, and on MSE for an audio dataset.

Complementary to those experimental works, Tran et al. (2021) conducted a theoretical study on the disparate impact of DPSGD. They investigated its causes and concluded that the primary factors include both data and model properties, such as input norms, distance to the decision boundary, clipping bound and privacy budget. While some of their analyses only hold for convex loss functions, they empirically validated their conclusions for non-convex cases. As a fairness measure, they relied on excessive risk (i.e., the difference between private and non-private expected loss). Along similar lines, Esipova et al. (2022) identified gradient misalignment (i.e., changes of the gradient direction rather than the magnitude) as the main cause for the disparate impact of DPSGD.

The role performance-based hyperparameter tuning has on the impact of DPSGD on fairness was first studied by de Oliveira et al. (2023). They pointed out that previous studies re-used hyperparameters that performed well for the non-private model, rather than tuning them specifically for DP.[2] By comparing tuned non-private models (trained with SGD) with both their DP counterparts trained with the same hyperparameters and a separately tuned DP model, de Oliveira et al. (2023) concluded that when specifically optimizing the hyperparameters for DP, DPSGD does not necessarily exhibit a disparate impact. In contrast to most previous work, however, they did not look at overall accuracy and accuracy equality but overall AUC-ROC, and AUC-ROC equality, demographic parity, equalized odds, and predictive parity. This work is the one most closely related to ours, although we deliberately made distinct experimental design decisions such as using binary groups instead of more fine-grained combinations of race and sex, adapting the tuned hyperparameters (see Section 3), and using the same privacy budget $\epsilon = 5$ for the untuned and tuned DPSGD models. We also replaced two of the tabular datasets with image datasets to diversify our experiments.

In addition to advancing the understanding of the disparate impact of DPSGD, several works developed mitigation strategies. The Xu et al. (2021) proposed to compute group-specific clipping norms, Tran et al. (2021) added fairness constraints to the empirical risk minimizer, and Zhang et al. (2021) suggested using early stopping to find the optimal trade-off between accuracy, fairness, and privacy. Based on their finding that gradient misalignment is the main cause of the disparate impact, Esipova et al. (2022) developed a variant of DPSGD that alleviates gradient direction changes by scaling the per-example gradients.

For a more detailed review of fairness under differential privacy, see Fioretto et al. (2022), which covers both decision and learning tasks, and also includes an overview of mitigation techniques.

---

[2]While hyperparameter tuning is common practice, Papernot & Steinke (2021) showed that hyperparameters can leak private information. This applies to both methods: Adopting hyperparameters from non-private models compromises the ability to account for the privacy budget, while tuning directly on DP models increases the effective privacy budget. Further elaboration on this issue can be found in the discussion (Section 7).

## 3 Methods

We chose six datasets that were previously used by similar studies (Farrand et al., 2020; Xu et al., 2021; Bagdasaryan et al., 2019; Tran et al., 2021; Esipova et al., 2022; de Oliveira et al., 2023): four tabular datasets (Adult (Becker & Kohavi, 1996), LSAC (Wightman, 1998), Compas (Angwin et al., 2016) and ACSEmployment Ding et al. (2021)) and two image datasets (CelebA (Liu et al., 2015) and MNIST (LeCun)). For all tabular datasets except ACSEmployment, the protected attribute is sex (as a binary attribute, i.e., male or female). For the Adult dataset, the task is income prediction (either $\leq$ 50k or $>$ 50k). For LSAC, it is the prediction of bar exam results (either fail/not attempted or pass). For Compas, it is re-arrest prediction (yes or no). For ACSEmployment the task is employment status prediction; as the protected attribute we chose vision difficulty due to its high imbalance between groups. CelebA consists of images of faces for which we followed Esipova et al. (2022) and performed gender classification (male or female) while wearing eyeglasses (yes or no) is defined as the protected attribute. For MNIST, an image dataset for digit classification (digits 0 to 9), we followed Esipova et al. (2022) and reduced the samples for class 8 by around 90%. We then compared classes 2 and 8 with each other. A detailed description of the datasets can be found in the Appendix (see Section A.1).

Following Esipova et al. (2022), we trained a neural network with 3 linear layers, where the hidden layer has 256 units, for the tabular datasets. For the image datasets, we chose a CNN with 2 convolutional layers with 3x3 kernels and 32 and 16 channels, respectively. All models were trained with 5-fold cross-validation, where hyperparameter selections were based on the mean performance on the validation folds, and final results are reported on the hold-out test set. For performance, we consider accuracy, AUC-ROC, and AUC-PR. For fairness, we use the notions introduced in Section 2.2 and report the difference of the corresponding measure between the two groups, i.e., accuracy/AUC-ROC/AUC-PR difference for evaluating performance equality, acceptance rate difference for evaluating demographic parity, precision difference for predictive parity, maximum of false positive and false negative error rate difference for equalized odds.

Unless otherwise specified, our experiments were conducted with a privacy budget of $\epsilon = 5$ and $\delta = 1e - 5$ and $\delta = 1e - 6$ for tabular and image datasets respectively.[3] We also tested different $\epsilon \in \{0.5, 1, 10\}$ for the three tabular datasets, but observed no unexpected deviations. In the case of hyperparameter tuning, we performed either grid search or random search over the following hyperparameter values:[4]

- Learning rate: [0.0001, 0.001, 0.01, 0.1]
- Batch size: [256, 512]
- Number of epochs: [5, 10, 20, 40]

- Activation function: [tanh, relu]
- Optimizer: [SGD, Adam]
- Clipping norm (for DPSGD): [0.01, 0.1, 1]

We performed grid search for the smaller tabular datasets (which results in 128 settings for non-DP, and 384 settings for DPSGD and DPSGD-Global-Adapt), and random search for the ACSEmployment dataset and the image datasets CelebA and MNIST (with 50 samples for non-DP, and the corresponding 150 samples for DP). For the sake of reproducibility (Semmelrock et al., 2025), our source code is based on Esipova et al. (2022) and is available via GitHub at `https://github.com/leakatharina24/Paper_DisparateImpactOfDPSGD`.

To focus on general hyperparameters and avoid that results are dominated by the choice of clipping norm, we consistently report outcomes using the clipping norm that yields the best overall performance. For comparison, results obtained with the worst-performing clipping norm are provided in the appendix.

Whenever we report negative or positive influences (i.e., for all tables and heatmaps in the main part of the paper), we base our conclusions on significance tests rather than direct mean comparison or threshold-based rules, ensuring that variability is accounted for when assessing differences between the models (see Appendix A.2 for more details).

In addition to standard DPSGD, we also investigated DPSGD-Global-Adapt. Instead of simply clipping per-example gradients that exceed the clipping norm, DPSGD-Global-Adapt uniformly scales all gradients

---

[3]These values were chosen as the image datasets are larger, therefore a lower $\delta$ is needed to follow the convention of choosing a $\delta$ smaller than the inverse of the dataset size.

[4]We adopted the hyperparameter values from de Oliveira et al. (2023) with the only difference that we keep model depth constant, do not include dropout, but additionally tune the number of epochs.

so they fall below the (adaptive) threshold. This consistent scaling helps prevent gradient misalignment and preserves directional information. We chose this particular method for mitigating the disparate impact of DPSGD because, similar to performance-based hyperparameter optimization, it is not necessary to know group information during training. Moreover, the method is simple to apply due to the openly accessible code and showed better results compared to Xu et al. (2021) and Tran et al. (2021) (both of which need access to the protected attribute during training). We also did not consider the early stopping method by Zhang et al. (2021) as it would require a public, non-private validation set.

## 4   Beyond DPSGD's Disparate Impact on Accuracy

As outlined in Section 2.3, most works studying the impact of DPSGD on fairness refer to its disparate impact on accuracy (Bagdasaryan et al., 2019; Farrand et al., 2020; Xu et al., 2021; Esipova et al., 2022; Tran et al., 2021).[5] Farrand et al. (2020) additionally investigate demographic parity and equal opportunity. The work from de Oliveira et al. (2023) is the first to refrain from using accuracy difference, investigating AUC-ROC difference, demographic parity, predictive parity, and equalized odds instead. They presumed that DPSGD has a disparate impact on these fairness metrics (without hyperparameter tuning) even though their experiments could not unambiguously verify this. Out of five datasets, only two exhibit significantly larger AUC-ROC differences for the untuned DP model compared to the non-private model. The results for acceptance rate difference, equalized odds difference, and precision difference are similarly ambiguous (see Table 1 in (de Oliveira et al., 2023) or our analysis of their results in Table 5 in the Appendix A.3).

With the objective of enhancing clarity in this matter, we conducted our own experiments (see detailed explanations in Section 3). Table 1 shows a concise overview of our results regarding the impact of DPSGD on a wider range of metrics than previously considered.[6] As expected, our results show a negative impact of DP on performance across all three performance metrics. They also demonstrate a disparate impact on performance metrics, although not for all datasets consistently - providing further evidence that the influence of DPSGD on different fairness metrics is highly dependent on the specific dataset. Notably, a disparate impact on one performance metric does not necessarily imply a disparate impact on another, e.g., for the Adult dataset, DPSGD has a negative effect on the accuracy difference but not on the AUC differences, for the LSAC dataset, only the AUC-PR difference is negatively impacted, and for the ACSEmployment dataset both AUC differences but not accuracy difference is degraded. For the other fairness metrics (acceptance rate difference, equalized odds difference, and precision difference), we also do not observe a consistent negative impact or consistent co-occurrence patterns.

> **Takeaways (RQ1)**
> The impact of DPSGD on fairness tends to depend not only on the dataset but also on the choice of metrics. Conclusions drawn from one metric do not necessarily apply to another - even within the same category of fairness metric, such as performance equality.

## 5   The Role of Hyperparameter Choice

After establishing that the disparate impact of DPSGD does not manifest consistently across metrics and that conclusions drawn from one metric does not necessarily apply to another, we now revisit the role hyperparameters play. de Oliveira et al. (2023) argued that DPSGD does not necessarily exhibit a disparate impact when specifically optimizing the hyperparameters for DP. Re-analyzing their results (see Table 6 in the Appendix), we show that while it is true that in some cases (e.g., for the ACS Inc. dataset) hyperparameter tuning mitigates the disparate impact of DPSGD, it does not seem to hold in general. For example, the disparate impact on AUC-ROC in the case of the Adult dataset is not improved by hyperparameter tuning.

---

[5]For practical reasons, the theoretical analyses of Tran et al. (2021) and Esipova et al. (2022) look at excessive risk (i.e., the difference between private and non-private risk functions or, in other words, the expected loss differences) but they both motivate it via accuracy and Esipova et al. (2022) additionally uses accuracy difference as a metric in their experiments.

[6]The table shows the results for the clipping norm that achieves the best overall performance (measured by accuracy, AUC-ROC, and AUC-PR, respectively). The conclusions, however, do not change even if the worst clipping norms are considered (see Table 7 in the Appendix). You can also find tables containing our full results in the Appendix A.6.

Table 1: Negative impact of DPSGD on the respective metrics. The crosses (✗) indicate significantly worse outcomes of the DPSGD model compared to the tuned SGD model, using the same hyperparameters and the clipping norm with the *best* overall performance. Acceptance rate and equalized odds are not applicable (N/A) metrics for MNIST, as the comparison is made between classes rather than groups. The precision difference is not defined (n.d.) when a model only predicts the negative class.

| | Adult | LSAC | Compas | ACSEmployment | CelebA | MNIST |
|---|---|---|---|---|---|---|
| Overall accuracy | ✗ | ✗ | ✗ | ✗ | ✗ | ✗ |
| Accuracy difference | ✗ | - | ✗ | - | ✗ | ✗ |
| Acceptance rate difference | ✗ | - | - | ✗ | - | N/A |
| Equalized odds difference | ✗ | - | - | ✗ | ✗ | N/A |
| Precision difference | - | - | n.d. | - | - | ✗ |
| Overall AUC-ROC | ✗ | ✗ | ✗ | ✗ | ✗ | ✗ |
| AUC-ROC difference | - | - | ✗ | ✗ | ✗ | ✗ |
| Acceptance rate difference | - | - | ✗ | ✗ | - | N/A |
| Equalized odds difference | - | - | ✗ | ✗ | ✗ | N/A |
| Precision difference | - | - | - | - | - | ✗ |
| Overall AUC-PR | ✗ | ✗ | ✗ | ✗ | ✗ | ✗ |
| AUC-PR difference | - | ✗ | ✗ | ✗ | ✗ | ✗ |
| Acceptance rate difference | - | - | - | ✗ | - | N/A |
| Equalized odds difference | - | - | - | ✗ | ✗ | N/A |
| Precision difference | - | - | - | - | - | ✗ |

Thus, a deeper analysis of hyperparameter tuning is still needed in the context of the disparate impact of DPSGD.

As a prefatory remark, it has been shown that the clipping norm is an important factor in the disparate impact of DPSGD (Bagdasaryan et al., 2019; Tran et al., 2021). This could lead one to the conclusion that tuning the clipping norm alone might already mitigate the disparate impact of DPSGD. de Oliveira et al. (2023) used a fixed clipping norm for their untuned DPSGD models. In contrast, we report the untuned DPSGD model *with their respective best-performing clipping norm*. The results for the worst-performing clipping norms can be found in Appendix Section A.5. We observed that only tuning the clipping norm does not considerably and consistently improve the negative impact of DPSGD. It also does not change our conclusions.

Table 2 shows our results on whether hyperparameter tuning improves the impact of DPSGD on performance and fairness. While performance-based hyperparameter tuning significantly improves the performance of DPSGD in all cases, the results are less clear for fairness. While in some settings, hyperparameter tuning mitigates disparities introduced by DP for part of the metrics, sometimes even eliminating the negative impact of DP altogether, it fails to improve fairness consistently. For the ACSEmployment dataset, tuning hyperparameters is unable to improve any fairness metric.

We have to keep in mind, however, that when we look at only the best-performing hyperparameter setting, we ignore the fact that similarly performing hyperparameter settings can exhibit considerably different (un)fairness levels (see Fig. 8 in the Appendix for an illustrative example of this circumstance). Thus, in the following, we will present the results of our experiments over the full range of tested hyperparameters. For those results, we will exclusively report accuracy and accuracy differences, given that the disparate impact on accuracy is the most consistently observed effect of DPSGD on fairness.

Figs. 1-6A show accuracy and accuracy difference over the different hyperparameter settings. The solid blue line represents the tuned SGD models plotted over the hyperparameter settings sorted by accuracy. The dash-dot green line depicts the corresponding DPSGD models using the same hyperparameter settings as the SGD model. The dashed orange line shows the tuned DPSGD models over the hyperparameters, sorted by their accuracy. This means that for one position on the x-axis, the solid blue and dash-dot green line share the same hyperparameter setting, while the dashed orange one does not. The heatmaps 1-6B summarize

Table 2: Improvements on the impact of DPSGD on the respective metrics through performance-based hyperparameter tuning. The checkmarks (✓) indicate significant improvements over the untuned DPSGD model (using the clipping norm with the *best* overall performance). The stars (*) mark results where the tuned DPSGD eliminates the disparate impact of DPSGD, i.e., the tuned DPSGD model performs similar or better than the tuned SGD model, while the untuned does not. Acceptance rate and equalized odds are not applicable (N/A) metrics for MNIST, as the comparison is made between classes rather than groups. The precision difference is not defined (n.d.) when a model only predicts the negative class.

| | Adult | LSAC | Compas | ACSEmployment | CelebA | MNIST |
|---|---|---|---|---|---|---|
| Overall accuracy | ✓ | ✓ | ✓ | ✓ | ✓ | ✓ |
| Accuracy difference | ✓ | ✓ | ✓⋆ | - | - | - |
| Acceptance rate difference | ✓⋆ | - | - | - | - | N/A |
| Equalized odds difference | ✓⋆ | - | - | - | - | N/A |
| Precision difference | - | ✓ | n.d. | - | - | - |
| Overall AUC-ROC | ✓ | ✓ | ✓⋆ | ✓ | ✓ | ✓ |
| AUC-ROC difference | - | - | ✓⋆ | - | ✓ | - |
| Acceptance rate difference | - | - | ✓⋆ | - | - | N/A |
| Equalized odds difference | - | - | ✓⋆ | - | - | N/A |
| Precision difference | - | - | - | - | - | - |
| Overall AUC-PR | ✓ | ✓ | ✓⋆ | ✓ | ✓ | ✓ |
| AUC-PR difference | - | ✓⋆ | ✓⋆ | - | ✓ | ✓ |
| Acceptance rate difference | ✓ | - | - | - | - | N/A |
| Equalized odds difference | ✓ | - | - | - | - | N/A |
| Precision difference | ✓ | - | - | - | - | ✓⋆ |

how often DPSGD achieves better/similar/worse performance and is fairer/similarly fair/unfairer than the SGD model with the same hyperparameters (i.e., it compares the solid blue and dash-dot green lines).

The first thing we can observe is that DPSGD does not have a negative impact on performance and/or fairness for all hyperparameter settings. While for most datasets, the percentage of settings for which DPSGD leads to worse and unfairer results constitutes the largest part, a considerable number of hyperparameter settings elicit DPSGD models that achieve similar or, in rare cases, even better results for at least one of the two measures.

By comparing the SGD models with their corresponding DPSGD version (i.e., the solid blue with the dash-dot green line), we can see that hyperparameter settings that achieve high accuracy for the SGD model may or may not perform well for the DPSGD model. That is to say, the dash-dot green line sometimes reaches high accuracy (for some datasets even similar values to the non-DP model) but exhibits strong fluctuations in the unfavorable direction.

Moreover, we can observe that, for accuracy difference, the dashed orange curve is smoother than the dash-dot green one, in particular for higher accuracy settings, and except for LSAC, where the two lines are more similar. This means that DP models with similar accuracy exhibit lower variations in accuracy differences than DP models where the non-DP counterparts achieve similar accuracy, leading to the conclusion that performance-based hyperparameter tuning of the DP model produces more reliable results than re-using well-performing hyperparameters from the non-private model. However, this does not mean that DPSGD achieves competitive performance and fairness compared to the non-DP model: For one, the DP model often does not reach non-DP performance (see Compas, LSAC, CelebA, MNIST). Secondly, high-performing DP models sometimes do not reach non-DP fairness levels (see CelebA and MNIST). Interestingly, our experiments suggest that multi-objective hyperparameter tuning that takes both performance and fairness into account (for example along the line of Cruz et al. (2021)) would not be able to considerably improve

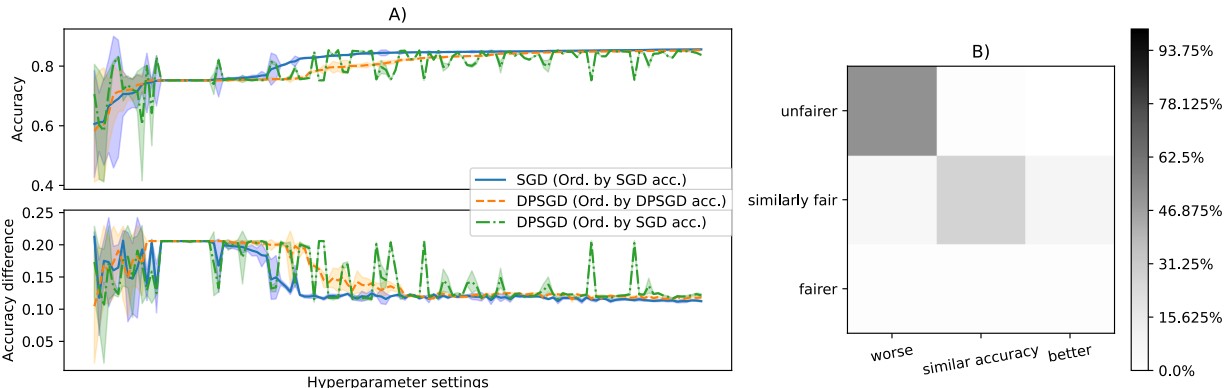

Figure 1: Results over all hyperparameter settings for the Adult dataset. A) shows accuracy and accuracy difference over all tested hyperparameter settings for the SGD and DPSGD models. Intervals shown correspond to $\pm 1$ standard deviation, reflecting variability across the 5 training runs. The results for the SGD model, represented by the solid blue line, are ordered by its accuracy. The dash-dot green line illustrates the DPSGD model with the same hyperparameters as the SGD model. The dashed orange line shows the results for the DPSGD model ordered by its own accuracy. Takeaway: As expected, hyperparameter settings that result in high accuracy for SGD do not necessarily do so for DPSGD. Interestingly, accuracy and accuracy difference are negatively correlated, i.e., hyperparameter settings that result in lower performance also result in lower fairness. B) summarizes how often DPSGD achieves better/similar/worse performance and is fairer/similarly fair/unfairer than the SGD model with the same hyperparameters. Takeaway: While for most hyperparameter settings DPSGD has a negative effect on both performance and fairness, there exist some settings for which DPSGD results in similar accuracy difference and similar or even better overall accuracy.

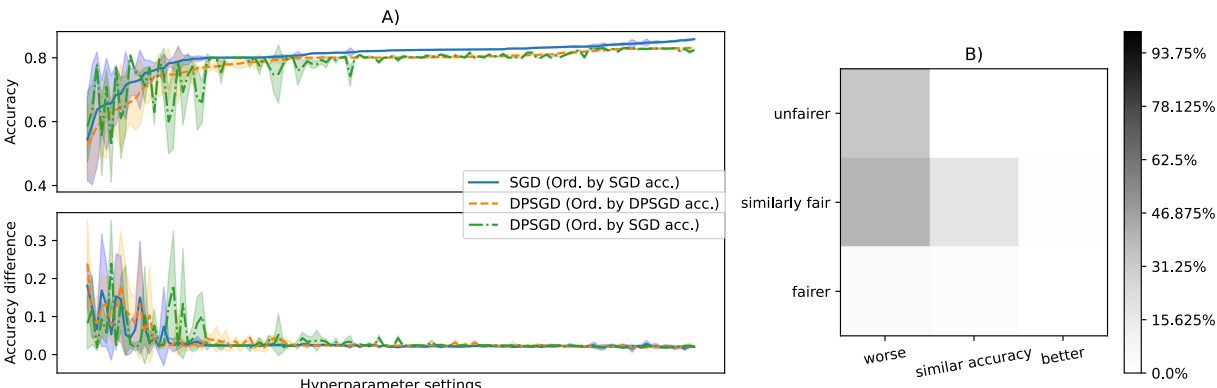

Figure 2: Results over all hyperparameter settings for the LSAC dataset (details explained in Fig. 1). Takeaway: For this dataset, DPSGD results in slightly worse accuracy but similar accuracy difference than SGD for most hyperparameter settings.

the performance-fairness trade-off.[7] While there are remaining fluctuations in the dashed orange line, they are small for high accuracy settings.

---

[7]Unlike for performance-based hyperparameter tuning, multi-objective hyperparameter tuning would require that the protected groups are known during training.

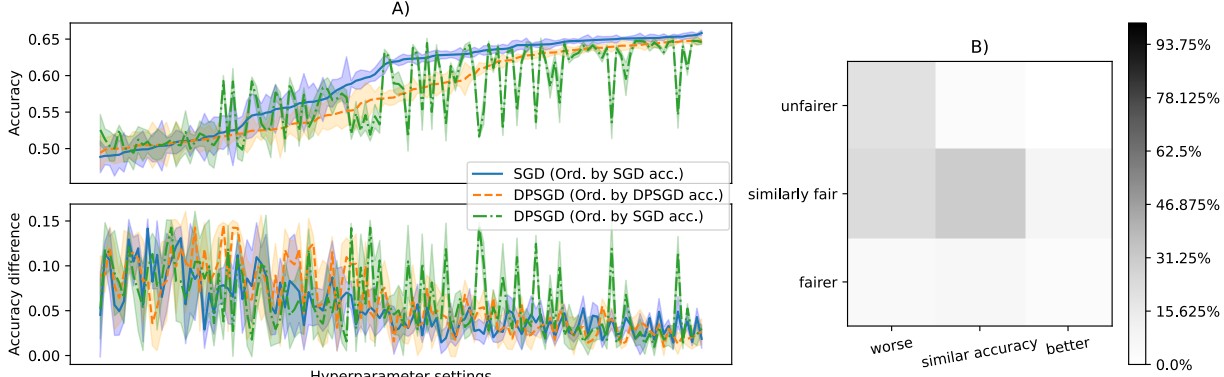

Figure 3: Results over all hyperparameter settings for the Compas dataset (details explained in Fig. 1). Takeaway: Choosing hyperparameters for DPSGD based on SGD accuracy leads to unpredictable accuracy and accuracy difference: While some hyperparameters work well for both, others exhibit considerably worse performance and fairness for DPSGD. In general, higher accuracy difference coincides with lower overall accuracy.

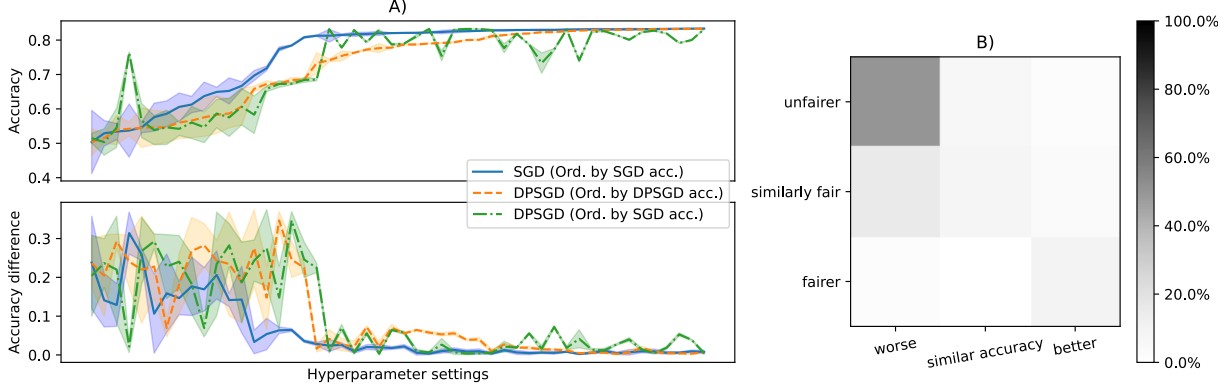

Figure 4: Results over all hyperparameter settings for the ACSEmployment dataset (details explained in Fig. 1). Takeaway: Choosing hyperparameters for DPSGD based on SGD accuracy leads to more unpredictable accuracy difference than tuning on DPSGD itself. For most hyperparameter settings DPSGD results in worse performance and increased unfairness.

---

**Takeaways (RQ2)**

DPSGD did not demonstrate a disparate impact on accuracy across all hyperparameter settings. Hyperparameter tuning of the DP model may not reliably result in competitive accuracy and accuracy difference compared to the tuned non-DP model, but generally yields more reliable results alongside higher accuracies than re-using the hyperparameters from the tuned non-DP model. Therefore, when training models with DPSGD, it appears to be beneficial to do hyperparameter tuning from a fairness point of view. Multi-objective hyperparameter optimization seems to offer limited potential to improve the performance-fairness trade-off further.

---

# 6 DPSGD-Global-Adapt and Hyperparameter Choice

The DPSGD variant DPSGD-Global-Adapt (Esipova et al., 2022) showed improved accuracy and accuracy differences on four datasets compared to DPSGD (and other mitigation methods). However, the authors only tested a fixed setting of hyperparameters, and did not perform separate hyperparameter optimization.

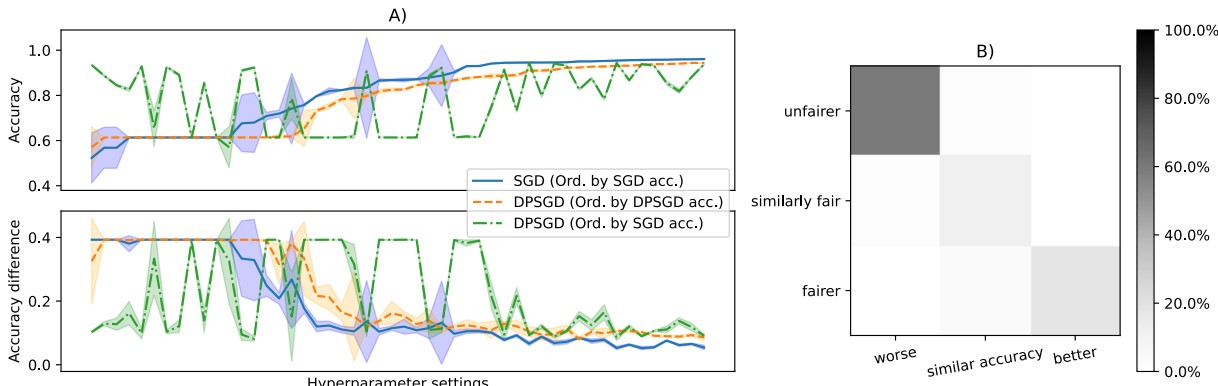

Figure 5: Results over all hyperparameter settings for the CelebA dataset (details explained in Fig. 1). Takeaway: Again, higher overall accuracy correlates with better fairness. Which hyperparameters work best for SGD and DPSGD respectively varies significantly, however, settings which yield high accuracy for SGD tend to be comparably stable when applied to DPSGD, but still can lead to considerably less performance and fairness.

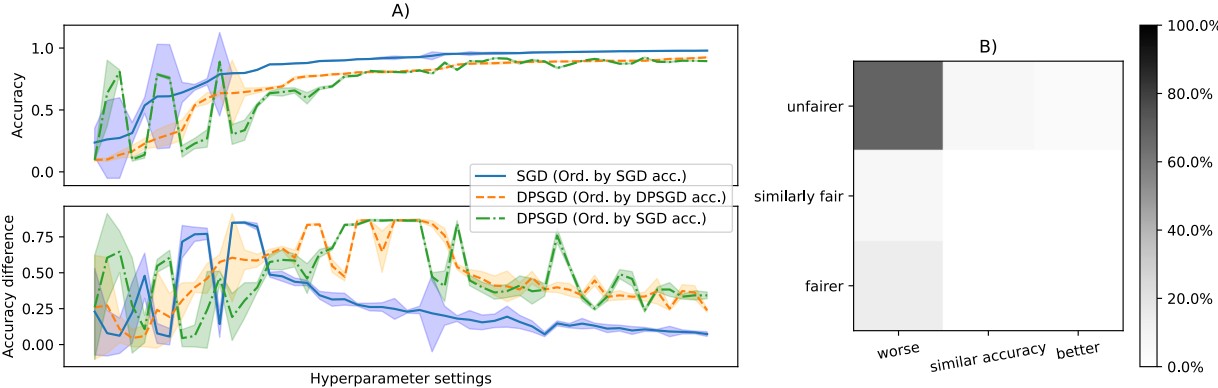

Figure 6: Results over all hyperparameter settings for the MNIST dataset (details explained in Fig. 1). Takeaway: DPSGD results in lower accuracy and higher accuracy difference for all hyperparameters except those that yield low performance for SGD.

We thus expand their analysis to answer the following questions: 1) Does DPSGD-Global-Adapt outperform DPSGD also over a wide range of hyperparameters? 2) If performance-based hyperparameter tuning is performed, does DPSGD-Global-Adapt outperform DPSGD?

The heatmaps in Fig. 7 illustrate the comparison between DPSGD and DPSGD-Global-Adapt across accuracy and accuracy difference, with each cell representing whether DPSGD-Global-Adapt performs worse, similar, or better than DPSGD for the respective metrics. Again, the results vary across datasets. For Adult, CelebA and ACSEmployment, DPSGD-Global-Adapt outperforms standard DPSGD for the majority of hyperparameter settings. For LSAC, the distribution is less positively skewed but still includes a few hyperparameter settings where DPSGD-Global-Adapt negatively impacts either performance or fairness compared to DPSGD. For Compas, DPSGD-Global-Adapt mainly performs similarly to DPSGD, sometimes decreasing performance. For MNIST, DPSGD-Global-Adapt mainly results in similar or improved performance but occasionally deteriorates fairness. We conclude that while DPSGD-Global-Adapt is not robustly superior to DPSGD over a wide range of hyperparameter settings, more often than not, it improves or at least does not worsen performance and fairness.

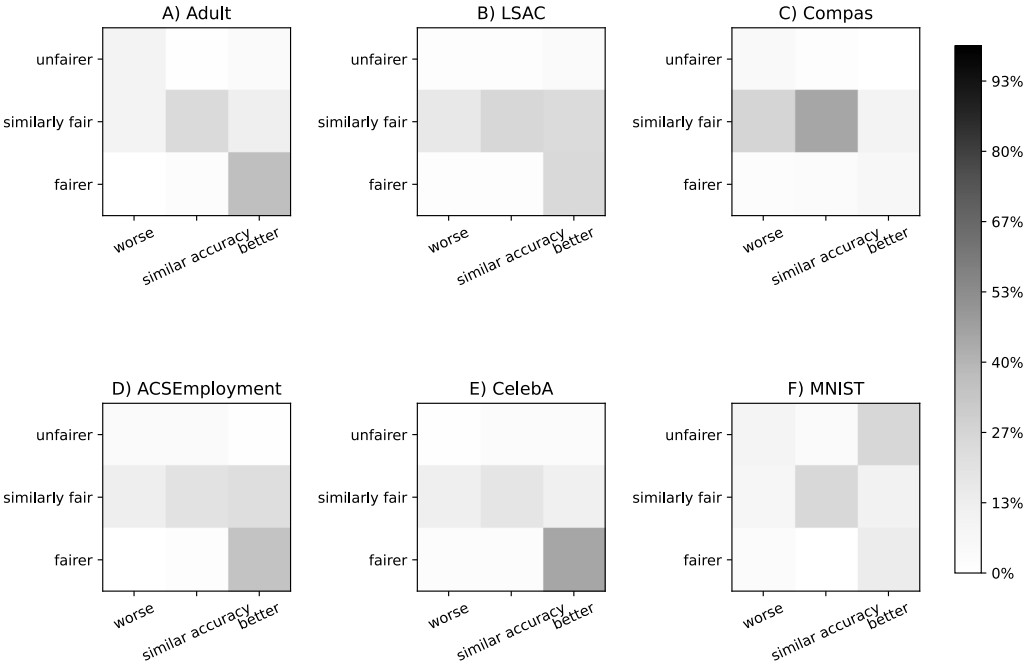

Figure 7: DPSGD-Global-Adapt compared to DPSGD over all hyperparameter. The heatmaps illustrate how often DPSGD-Global-Adapt achieves better/similar/worse performance and is fairer/similarly fair/unfairer than the standard DPSGD model with the same hyperparameters.

In Table 3 we report the cases in which DPSGD-Global-Adapt improves on all considered metrics compared to DPSGD, for both the untuned and tuned setting. The results do not allow a definitive answer to the question of whether DPSGD-Global-Adapt outperforms DPSGD. For some datasets and metrics, it does, but for most, it does not. There is also no consistency between untuned and tuned settings in terms of for which datasets and metrics DPSGD-Global-Adapt performs better. This weakens the usefulness of DPSGD-Global-Adapt. However, in most cases, DPSGD-Global-Adapt also does not negatively impact the results which still makes it eligible to be a reasonable alternative to DPSGD.

> **Takeaways (RQ3)**
> The analysis of different hyperparameter settings reveals that DPSGD-Global-Adapt does not robustly achieve better fairness than standard DPSGD. This finding persists even when hyperparameters are optimized for performance. Considering that DPSGD-Global-Adapt seldom compromises performance and fairness compared to DPSGD, it remains a reasonable alternative. However, it may not sufficiently address the adverse effects of DP on fairness.

# 7 Limitations and Discussion

Following the line of previous studies, this paper draws a comparison across models to isolate the algorithmic influence of DP. However, our results show that even in this setting, the data dependence cannot be disregarded, as data and algorithmic properties interrelate. Moreover, we emphasize that while ensuring the implementation of privacy-preserving technologies does not inadvertently increase unfairness is a crucial first step, the ultimate goal is to train private models that uphold fairness overall. Thus, we consider it

Table 3: Improvements of DPSGD-Global-Adapt on the respective metrics compared to DPSGD for both the untuned and tuned setting. The checkmarks indicate significant improvements over standard DPSGD, using the respective clipping norm with the *best* overall performance. Acceptance rate and equalized odds are not applicable (N/A) metrics for MNIST, as the comparison is made between classes rather than groups. The precision difference is not defined (n.d.) when a model only predicts the negative class.

| | Adult | | LSAC | | Compas | | ACSEmployment | | CelebA | | MNIST | |
|---|---|---|---|---|---|---|---|---|---|---|---|---|
| Tuned | no | yes | no | yes | no | yes | no | yes | no | yes | no | yes |
| Overall accuracy | - | ✓ | - | - | - | - | - | - | ✓ | ✓ | - | ✓ |
| Accuracy difference | ✓ | - | - | - | - | - | - | - | ✓ | ✓ | - | ✓ |
| Acceptance rate difference | - | - | - | - | - | - | - | - | - | - | N/A | N/A |
| Equalized odds difference | - | - | - | - | - | - | - | - | ✓ | - | N/A | N/A |
| Precision difference | - | - | - | - | n.d. | - | - | - | - | ✓ | - | ✓ |
| Overall AUC-ROC | ✓ | ✓ | ✓ | - | - | - | ✓ | ✓ | ✓ | ✓ | ✓ | - |
| AUC-ROC difference | - | - | ✓ | - | - | - | - | - | ✓ | - | - | ✓ |
| Acceptance rate difference | - | - | - | - | - | - | - | - | - | - | N/A | N/A |
| Equalized odds difference | - | - | - | - | - | - | - | - | - | - | N/A | N/A |
| Precision difference | - | - | - | - | - | - | - | - | - | - | - | ✓ |
| Overall AUC-PR | ✓ | ✓ | ✓ | ✓ | - | - | ✓ | ✓ | ✓ | ✓ | - | - |
| AUC-PR difference | - | - | ✓ | - | - | ✓ | - | - | ✓ | - | - | - |
| Acceptance rate difference | - | - | - | - | - | ✓ | - | - | - | - | N/A | N/A |
| Equalized odds difference | - | - | - | - | - | ✓ | - | - | - | - | N/A | N/A |
| Precision difference | - | - | - | - | - | - | - | - | - | - | - | - |

imperative to examine the broader context of fairness, even though the cross-model comparison approach can yield valuable insights.

Our results show that the impact of DPSGD highly depends on the used fairness metric, and that a negative impact on one metric does not necessarily result in a negative impact of another metric. That fairness metrics capture diverse fairness notions and, thus, lead to substantially different outcomes is a well-known challenge in fairness research (see e.g., Verma & Rubin (2018)). Our work shows that these challenges extend to privacy-fairness trade-offs, where different fairness metrics are affected differently by DPSGD. Moreover, our results demonstrate that the algorithmic influence of DPSGD (as well as DPSGD-Global-Adapt) is interdependent with the influence of hyperparameters. This aligns with previous observations that using DPSGD instead of standard SGD necessitates different architecture and hyperparameter choices to achieve high utility Papernot et al. (2021), and shows that these findings extend to fairness.

As already briefly mentioned in Section 2.3, Papernot & Steinke (2021) have drawn attention to the potential information leakage from optimized hyperparameters. The resulting trade-off should be carefully weighed when training differentially private ML models. Tuning hyperparameters on differentially private models - as done in this study - in contrast to optimizing them on non-private models intuitively reduces the potential privacy leakage and avoids utility losses that may arise due to differences in model behavior between non-private and private training Papernot et al. (2021). However, to ensure a theoretically sound privacy guarantee that both minimizes and accounts for the privacy loss elicited by hyperparameter tuning, it would be necessary to apply differentially private hyperparameter tuning (e.g., private random search (Liu & Talwar, 2019; Papernot & Steinke, 2021)). Differentially private hyperparameter tuning is still "in its infancy" (Ponomareva et al., 2023) and additional effort is needed to develop methods that are compatible with standard machine learning practices such as cross-validation. Moreover, empirical analyses of the privacy leakage through hyperparameter tuning suggest a significantly lower privacy cost than the current theoretical bounds, and hardly any leakage in practical settings, where the adversary does not have control over, e.g., the validation process (Xiang et al., 2024). We also show that similarly performing hyperparameter settings

for DP models result in similar fairness levels. Therefore, even if we had used private hyperparameter tuning, which reduces the chance of finding the best setting, we would still expect our results to remain similar.

## 8 Conclusions and Outlook

In this study, we investigated the role of metric and hyperparameter choice on the performance and fairness of DPSGD. Our findings show that DPSGD's disparate impact on one metric does not necessarily imply that it also has a disparate impact on another metric - even for metrics from the same category, e.g., performance metrics such as accuracy, AUC-ROC, and AUC-PR. Moreover, we demonstrate that the impact of DPSGD on fairness cannot be assumed to be consistent across a wide range of hyperparameter settings. We provide evidence that performance-based hyperparameter tuning is not a reliable method to achieve performance and fairness levels similar to non-private models, but conclude that it can yield improvements compared to re-using well-performing hyperparameter settings from non-private models. DPSGD-Global-Adapt, a variant of DPSGD proposed to mitigate its disparate impact on accuracy, does not demonstrate significant improvements in fairness compared to standard DPSGD in our experiments when hyperparameters are varied or tuned. This suggests that existing methods remain insufficient for reliably achieving strong performance–fairness trade-offs.

In future studies, it would be advisable to give careful consideration to the choice of metrics and exercise caution when generalizing findings from one metric to another. Moreover, more research is needed on data properties and their effect on fairness, also in studies that investigate the interplays between fairness and privacy. While our results suggest that when training private and fair ML models hyperparameters should be optimized directly on differentially private models rather than re-using those from non-private models, we caution that any hyperparameter tuning entails additional privacy leakage. Developing effective private hyperparameter tuning methods therefore remains an important direction for future research.

### Acknowledgments

This research was supported by the project *PRO'k'RESS* managed by the Austrian Research Promotion Agency FFG (grant number 60155996). Know Center Research GmbH is a COMET competence center that is financed by the Austrian Federal Ministry of Innovation, Mobility and Infrastructure (BMIMI), the Austrian Federal Ministry of Economy, Energy and Tourism (BMWET), the Province of Styria, the Steirische Wirtschaftsförderungsgesellschaft m.b.H. (SFG), the Vienna business agency and the Standortagentur Tirol. The COMET programme is managed by the Austrian Research Promotion Agency FFG.

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

# A  Appendix

## A.1  Details about datasets

Table 4 shows the class and group distributions for all six datasets. MNIST is the only dataset where the class distributions in the test set differ from those in the training set (i.e., classes in the test set are (more or less) equally distributed). For preprocessing the tabular datasets, we one-hot encoded the categorical features and standardized all features.

Table 4: Dataset class and group distributions. For Adult, LSAC and Compas, group 1 refers to female, group 0 to male. For the ACSEmployment, group 1 refers to no vision difficulty and group 0 to vision difficulty. For the Adult dataset, class 1 are incomes $> 50k$; for the LSAC dataset, class 1 are passed exams; for the Compas dataset, class 1 are re-arrests, and for ACSEmployment, class 1 is employment. For the CelebA dataset, class 1 is male, class 0 is female, and group 1 refers to individuals wearing eyeglasses. For MNIST, the digits 2 and 8 are compared.

| Dataset | Class imbalance | Group imbalance | Class imbalance in group 1 | Class imbalance in group 0 |
|---|---|---|---|---|
| Adult (1:0) | 25% : 75% | 32% : 68% | 12% : 88% | 31% : 69% |
| LSAC (1:0) | 80% : 20% | 44% : 56% | 80% : 20% | 80% : 20% |
| Compas (1:0) | 48% : 52% | 18% : 81% | 29% : 61% | 50% : 50% |
| ACSEmployment | 46% : 54% | 98% : 2% | 99% : 1% | 96% : 4% |
| CelebA (1:0) | 42% : 58% | 7% : 93% | 80% : 20% | 39% : 61% |
| MNIST (2:8) | 11% : 0.9% | 11% : 0.9% | - | - |

## A.2 Significance test

To compare two models based on a specific metric, we use 2-sample one-sided Welch's t-tests, with means $\mu_1$ and $\mu_2$, standard deviations $s_1$ and $s_2$ and sample sizes $n_1$ and $n_2$ (both 5 in our experiments, as we perform 5-fold cross-validation).

The t-statistic is computed with

$$t = \frac{\mu_1 - \mu_2}{s} \tag{2}$$

where

$$s = \sqrt{\frac{s_1^2}{n_1} + \frac{s_2^2}{n_1}} \tag{3}$$

The degrees of freedom (d.f.) are calculated using the following formula:

$$d.f. = \frac{(\frac{s_1^2}{n_1} + \frac{s_2^2}{n_2})^2}{\frac{(\frac{s_1^2}{n_1})^2}{n_1-1} + \frac{(\frac{s_2^2}{n_2})^2}{n_2-1}} \tag{4}$$

We reject our null hypotheses $H_0$: $\mu_1 \geq \mu_2$ or $H_0$: $\mu_1 \leq \mu_2$ if the p-value is below 0.05 and the t-statistic is negative or positive, respectively. We do not adjust for multiple comparisons to avoid overly conservative adjustments that could increase the risk of overlooking meaningful effect.

## A.3 Analysis of de Oliveira et al. (2023)

Table 5 and 6 show our analyses of Table 1 in (de Oliveira et al., 2023). As with our own tables, we used Welch's t-tests to determine if the influences are significant.

## A.4 Example of fairness variations between similarly performing hyperparameter settings

Fig. 8 shows the SGD and DPSGD models for the hyperparameter settings achieving the 5% best accuracies, and the untuned DPSGD models using the same hyperparameters as the 5% best SGD models. One can see, that similarly performing hyperparameter settings can exhibit considerably different (un)fairness levels.

Table 5: Analysis of Table 1 in (de Oliveira et al., 2023): Crosses (✗) indicate a negative impact of DPSGD on the respective metrics.

| | ACS Emp. | ACS Inc. | LSAC | Adult | Compas |
|---|---|---|---|---|---|
| Overall AUC-ROC | ✗ | ✗ | ✗ | ✗ | ✗ |
| AUC-ROC difference | - | ✗ | - | ✗ | - |
| Acceptance rate difference | - | ✗ | - | ✗ | - |
| Equalized odds difference | - | ✗ | - | - | - |
| Precision difference | - | ✗ | ✗ | - | - |

Table 6: Analysis of Table 1 in (de Oliveira et al., 2023): Checkmarks (✓) indicate improvements through hyperparameter tuning on the respective metrics. Stars (*) indicate instances, where the hyperparameter tuning eliminates the negative impact of DPSGD, i.e., the tuned DPSGD model performs similar or better than the tuned SGD model, while the untuned does not.

| | ACS Emp. | ACS Inc. | LSAC | Adult | Compas |
|---|---|---|---|---|---|
| Overall AUC-ROC | ✓ | ✓ | ✓ | ✓ | ✓⋆ |
| AUC-ROC difference | ✓ | ✓⋆ | - | - | ✓ |
| Acceptance rate difference | ✓ | ✓⋆ | - | ✓⋆ | - |
| Equalized odds difference | ✓ | ✓⋆ | ✓ | - | - |
| Precision difference | ✓ | ✓⋆ | - | - | - |

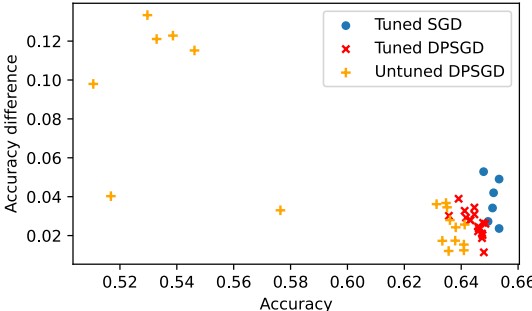

Figure 8: An example of the variations between hyperparameter settings based on the Compas dataset. It shows the SGD and DPSGD models with respective 5% best hyperparameter settings, and untuned DPSGD models using the same hyperparameters as the 5% best SGD models.

## A.5 Results for worst clipping norm

Tables 7, 8 and 9 show the equivalent to Tables 1, 2 and 3 but with the DPSGD model with the worst performing clipping norm instead of the best. Similarly, Figs. 9-14 show the equivalent of Figs. 1-6, and Fig. 15 the equivalent of Fig. 7.

## A.6 Full results

Tables 10-14 show the detailed results on which Tables 1, 2 and 3 (as well as their counterparts with worst clipping norms) are based on.

Table 7: Negative impact of DPSGD on the respective metrics. The crosses (✗) indicate significantly worse outcomes of the DPSGD model compared to the tuned SGD model, using the same hyperparameters and the clipping norm with the *worst* overall performance. Acceptance rate and equalized odds are not applicable (N/A) metrics for MNIST, as the comparison is made between classes rather than groups. The precision difference is not defined (n.d.) when a model only predicts the negative class.

|  | Adult | LSAC | Compas | ACSEmployment | CelebA | MNIST |
|---|---|---|---|---|---|---|
| Overall accuracy | ✗ | ✗ | ✗ | ✗ | ✗ | ✗ |
| Accuracy difference | ✗ | - | ✗ | ✗ | ✗ | ✗ |
| Acceptance rate difference | - | - | - | - | - | N/A |
| Equalized odds difference | - | - | - | - | ✗ | N/A |
| Precision difference | - | - | n.d. | ✗ | - | ✗ |
| Overall AUC-ROC | ✗ | ✗ | ✗ | ✗ | ✗ | ✗ |
| AUC-ROC difference | - | ✗ | ✗ | ✗ | ✗ | ✗ |
| Acceptance rate difference | - | - | - | - | - | N/A |
| Equalized odds difference | - | - | - | - | ✗ | N/A |
| Precision difference | n.d. | - | ✗ | ✗ | - | ✗ |
| Overall AUC-PR | ✗ | ✗ | ✗ | ✗ | ✗ | ✗ |
| AUC-PR difference | - | ✗ | - | ✗ | ✗ | ✗ |
| Acceptance rate difference | - | - | - | - | - | N/A |
| Equalized odds difference | - | - | - | - | ✗ | N/A |
| Precision difference | - | - | ✗ | ✗ | - | ✗ |

Table 8: Improvements on the impact of DPSGD on the respective metrics through performance-based hyperparameter tuning. The checkmarks (✓) indicate significant improvements over the untuned DPSGD model (using the clipping norm with the *worst* overall performance). The stars (*) mark results where the tuned DPSGD eliminates the disparate impact of DPSGD, i.e., the tuned DPSGD model performs similar or better than the tuned SGD model, while the untuned does not. Acceptance rate and equalized odds are not applicable (N/A) metrics for MNIST, as the comparison is made between classes rather than groups. The precision difference is not defined (n.d.) when a model only predicts the negative class.

|  | Adult | LSAC | Compas | ACSEmployment | CelebA | MNIST |
|---|---|---|---|---|---|---|
| Overall accuracy | ✓ | ✓ | ✓ | ✓ | ✓ | ✓ |
| Accuracy difference | ✓ | - | ✓⋆ | ✓ | - | - |
| Acceptance rate difference | - | - | - | - | - | N/A |
| Equalized odds difference | - | - | - | - | - | N/A |
| Precision difference | ✓ | - | n.d. | ✓⋆ | - | - |
| Overall AUC-ROC | ✓ | ✓ | ✓⋆ | ✓ | ✓ | ✓ |
| AUC-ROC difference | - | - | ✓⋆ | - | ✓ | - |
| Acceptance rate difference | - | - | - | - | - | N/A |
| Equalized odds difference | - | - | - | - | - | N/A |
| Precision difference | n.d. | - | ✓⋆ | ✓⋆ | - | - |
| Overall AUC-PR | ✓ | ✓ | ✓⋆ | ✓ | ✓ | ✓ |
| AUC-PR difference | - | ✓⋆ |  | - | ✓ | ✓ |
| Acceptance rate difference | - | - | - | - | - | N/A |
| Equalized odds difference | ✓ | - | - | - | - | N/A |
| Precision difference | ✓ | - | - | ✓⋆ | - | ✓⋆ |

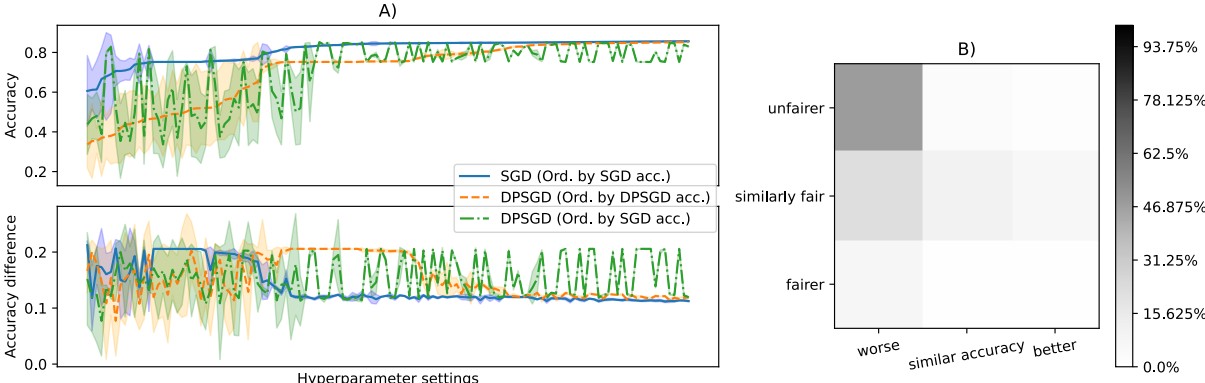

Figure 9: Results over all hyperparameter settings for the Adult dataset, using the clipping norms with the *worst* accuracy. A) shows accuracy and accuracy difference over all tested hyperparameter settings for the SGD and DPSGD models. Intervals shown correspond to ±1 standard deviation, reflecting variability across the 5 training runs. The results for the SGD model, represented by the solid blue line, are ordered by its accuracy. The dash-dot green line illustrates the DPSGD model with the same hyperparameters as the SGD model. The dashed orange line shows the results for the DPSGD model ordered by its own accuracy. Takeaway: As expected, hyperparameter settings that result in high accuracy for SGD do not necessarily do so for DPSGD. Interestingly, accuracy and accuracy difference are negatively correlated, i.e., hyperparameter settings that result in lower performance also result in lower fairness. B) summarizes how often DPSGD achieves better/similar/worse performance and is fairer/similarly fair/unfairer than the SGD model with the same hyperparameters. Takeaway: While for most hyperparameter settings DPSGD has a negative effect on both performance and fairness, there exist some settings for which DPSGD results in similar accuracy difference and similar or even better overall accuracy.

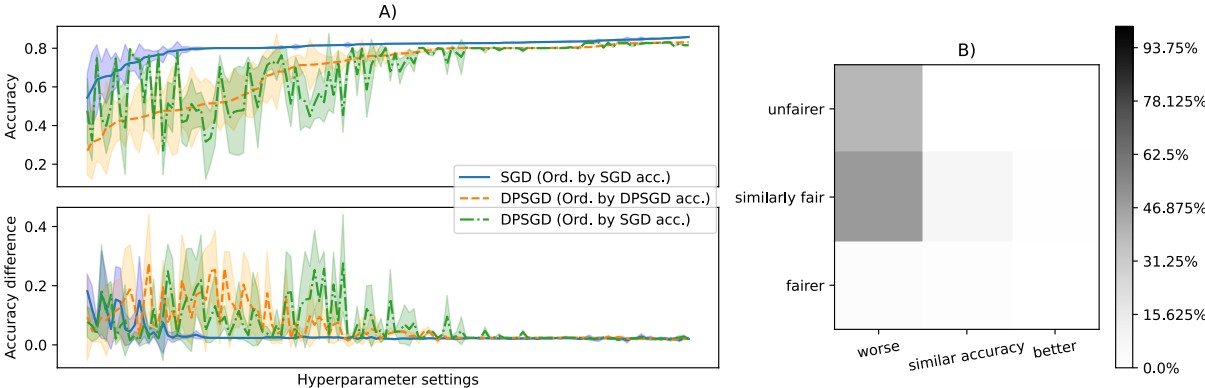

Figure 10: Results over all hyperparameter settings for the LSAC dataset, using the clipping norms with the *worst* accuracy (details explained in Fig. 9). Takeaway: For this dataset, DPSGD results in worse accuracy but similar accuracy difference than SGD for most hyperparameter settings.

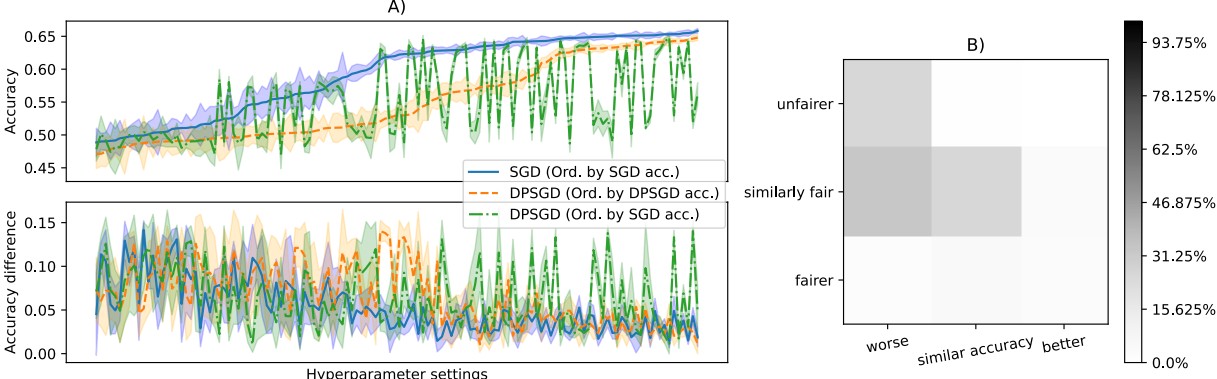

Figure 11: Results over all hyperparameter settings for the Compas dataset, using the clipping norms with the *worst* accuracy (details explained in Fig. 9). Takeaway: Choosing hyperparameters for DPSGD based on SGD accuracy leads to unpredictable accuracy and accuracy difference: While some hyperparameters work well for both, others exhibit considerably worse performance and fairness for DPSGD. In general, higher accuracy difference coincides with lower overall accuracy.

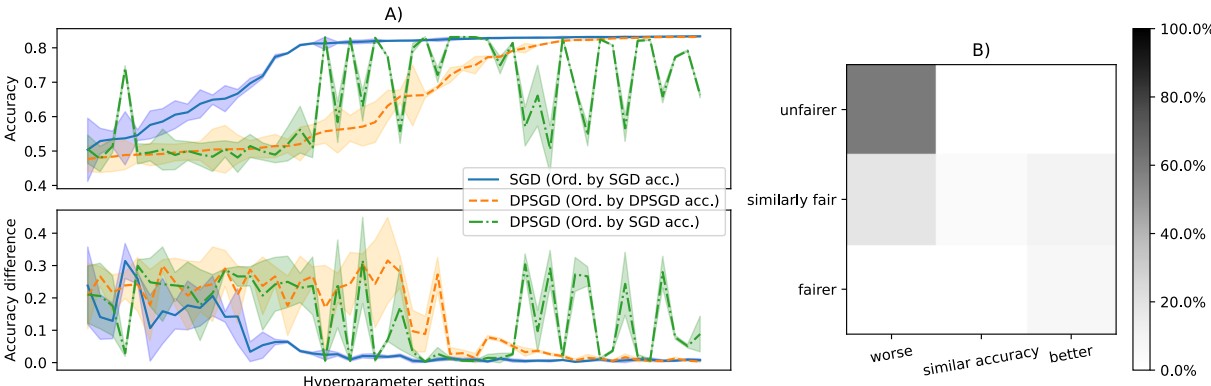

Figure 12: Results over all hyperparameter settings for the ACSEmployment dataset, using the clipping norms with the *worst* accuracy (details explained in Fig. 9). Takeaway: Choosing hyperparameters for DPSGD based on SGD accuracy leads to more unpredictable accuracy difference than tuning on DPSGD itself. For most hyperparameter settings DPSGD results in worse performance and increased unfairness.

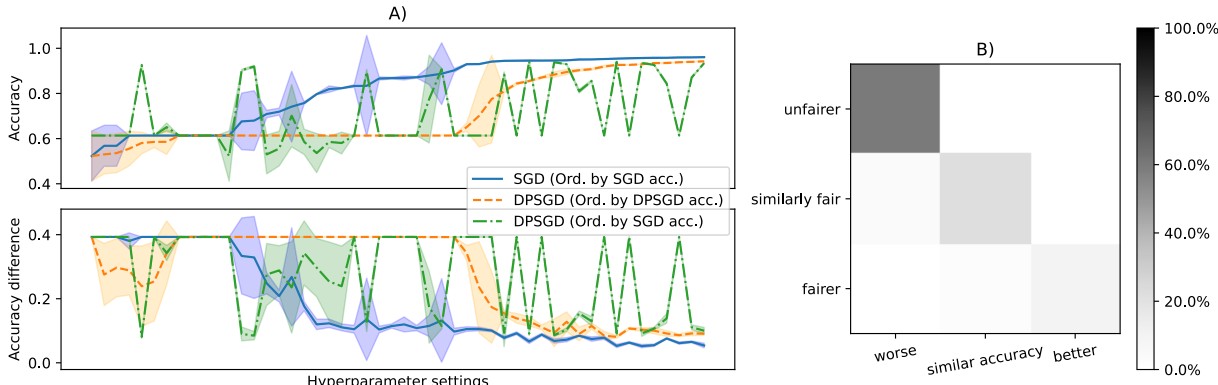

Figure 13: Results over all hyperparameter settings for the CelebA dataset, using the clipping norms with the *worst* accuracy (details explained in Fig. 9). Takeaway: Choosing hyperparameters for DPSGD based on SGD accuracy leads to unpredictable accuracy and accuracy difference: While some hyperparameters work well for both SGD and DPSGD, others exhibit considerably worse performance and fairness for DPSGD. Again, higher overall accuracy correlates with lower accuracy difference.

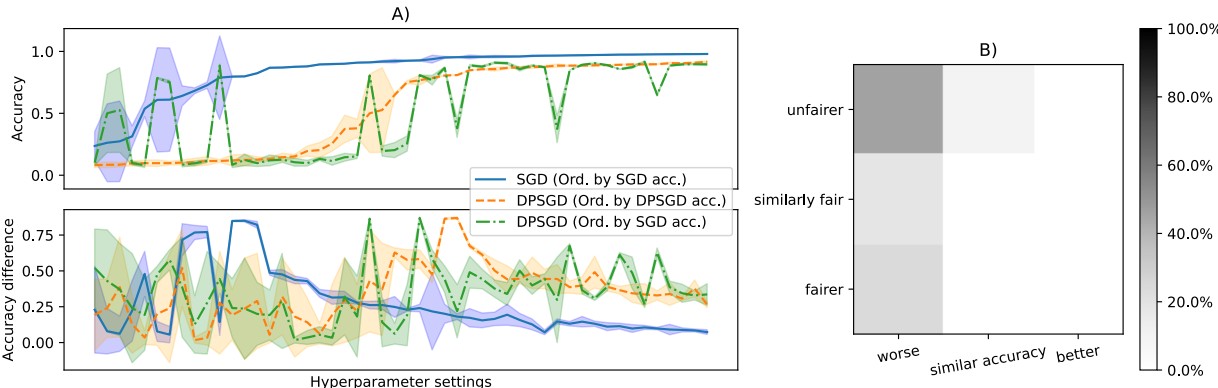

Figure 14: Results over all hyperparameter settings for the MNIST dataset, using the clipping norms with the *worst* accuracy (details explained in Fig. 9). Takeaway: DPSGD results in lower accuracy and higher accuracy difference for all hyperparameters except those that yield low performance for SGD.

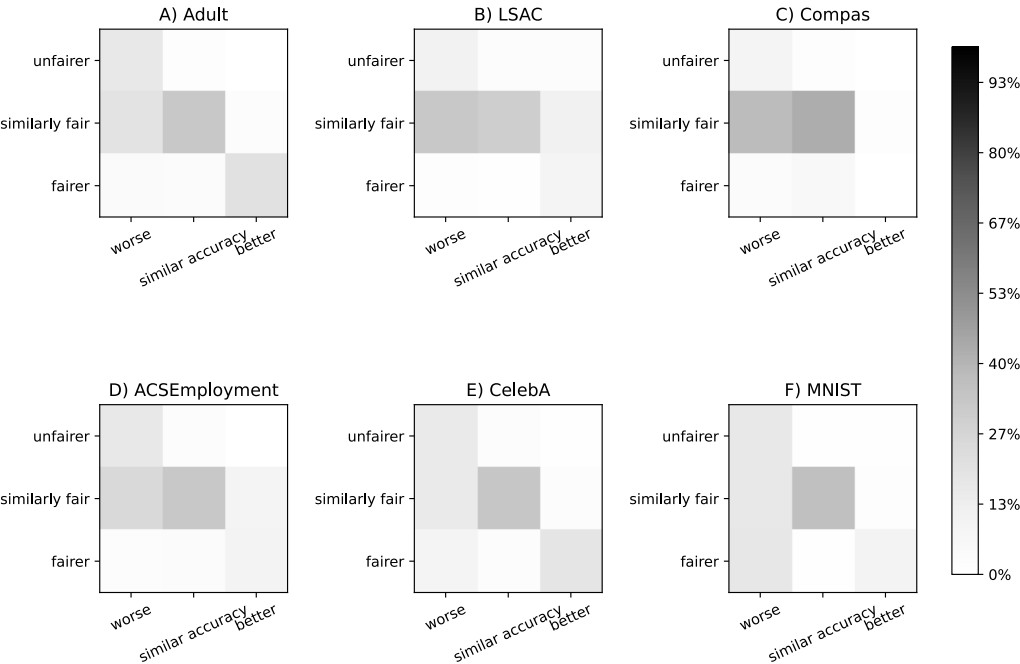

Figure 15: DPSGD-Global-Adapt compared to DPSGD over all hyperparameter, using the clipping norms with the *worst* accuracy. The heatmaps illustrate how often DPSGD-Global-Adapt achieves better/similar/worse performance and is fairer/similarly fair/unfairer than the standard DPSGD model with the same hyperparameters.

Table 9: Improvements of DPSGD-Global-Adapt on the respective metrics compared to DPSGD for both the untuned and tuned setting. The checkmarks (✓) indicate significant improvements over standard DPSGD, using the respective clipping norm with the *worst* overall performance. Acceptance rate and equalized odds are not applicable (N/A) metrics for MNIST, as the comparison is made between classes rather than groups. The precision difference is not defined (n.d.) when a model only predicts the negative class.

| | Adult | | LSAC | | Compas | | ACSEmployment | | CelebA | | MNIST | |
|---|---|---|---|---|---|---|---|---|---|---|---|---|
| Tuned | no | yes | no | yes | no | yes | no | yes | no | yes | no | yes |
| Overall accuracy | ✓ | ✓ | ✓ | - | ✓ | - | ✓ | - | ✓ | ✓ | - | ✓ |
| Accuracy difference | ✓ | - | - | - | - | - | - | - | ✓ | ✓ | - | ✓ |
| Acceptance rate difference | - | - | - | - | - | - | - | - | - | - | N/A | N/A |
| Equalized odds difference | - | - | - | - | - | - | - | - | - | - | N/A | N/A |
| Precision difference | ✓ | - | - | - | n.d. | - | - | - | - | ✓ | - | ✓ |
| Overall AUC-ROC | ✓ | ✓ | ✓ | - | - | - | - | ✓ | ✓ | ✓ | ✓ | - |
| AUC-ROC difference | - | - | ✓ | - | ✓ | - | - | - | ✓ | - | - | ✓ |
| Acceptance rate difference | - | - | - | - | ✓ | - | - | - | - | - | N/A | N/A |
| Equalized odds difference | - | - | - | - | ✓ | - | - | - | - | - | N/A | N/A |
| Precision difference | n.d. | - | - | - | - | - | - | - | - | - | - | ✓ |
| Overall AUC-PR | ✓ | ✓ | ✓ | ✓ | - | - | - | ✓ | ✓ | ✓ | - | - |
| AUC-PR difference | - | - | ✓ | - | - | ✓ | - | - | ✓ | - | - | - |
| Acceptance rate difference | - | - | - | - | - | ✓ | - | - | - | - | N/A | N/A |
| Equalized odds difference | - | - | - | - | - | ✓ | - | - | - | - | N/A | N/A |
| Precision difference | ✓ | - | - | - | - | - | - | - | - | - | - | - |

Table 10: Detailed results for the Adult dataset. Best/worst C denotes the clipping norm with the respective highest/lowest performance.

| | Tuned SGD | Untuned DPSGD (best C) | Untuned DPSGD (worst C) | Tuned DPSGD | Untuned DPSGD-G.-A. (best C) | Untuned DPSGD-G.-A. (worst C) | Tuned DPSGD-G.-A. |
|---|---|---|---|---|---|---|---|
| Overall accuracy | 0.8556 ± 0.0013 | 0.8491 ± 0.0005 | 0.8471 ± 0.0005 | 0.8532 ± 0.001 | 0.8498 ± 0.001 | 0.8499 ± 0.0004 | 0.8557 ± 0.0014 |
| Accuracy difference | 0.1126 ± 0.0027 | 0.1196 ± 0.0009 | 0.1236 ± 0.0005 | 0.1177 ± 0.0015 | 0.1183 ± 0.0007 | 0.1189 ± 0.0006 | 0.1145 ± 0.0016 |
| Acceptance rate difference | 0.1713 ± 0.0088 | 0.1832 ± 0.0063 | 0.1678 ± 0.0023 | 0.1682 ± 0.0044 | 0.1819 ± 0.0071 | 0.1907 ± 0.0057 | 0.1782 ± 0.007 |
| Equalized odds difference | 0.0657 ± 0.0068 | 0.0823 ± 0.0061 | 0.0699 ± 0.0047 | 0.0653 ± 0.0027 | 0.078 ± 0.0062 | 0.0895 ± 0.0076 | 0.0698 ± 0.0046 |
| Precision difference | 0.0206 ± 0.0084 | 0.0129 ± 0.0062 | 0.0222 ± 0.0068 | 0.0088 ± 0.0047 | 0.0092 ± 0.0061 | 0.0127 ± 0.0056 | 0.006 ± 0.0044 |
| Overall AUC-ROC | 0.9118 ± 0.0002 | 0.9077 ± 0.0003 | 0.8144 ± 0.0168 | 0.9096 ± 0.0002 | 0.91 ± 0.0003 | 0.837 ± 0.0095 | 0.911 ± 0.0003 |
| AUC-ROC difference | 0.0483 ± 0.0014 | 0.047 ± 0.0011 | 0.0207 ± 0.0137 | 0.0466 ± 0.0015 | 0.0472 ± 0.0004 | 0.0291 ± 0.0146 | 0.0476 ± 0.0008 |
| Acceptance rate difference | 0.1798 ± 0.0091 | 0.1731 ± 0.0036 | 0.0 ± 0.0 | 0.1736 ± 0.0045 | 0.1882 ± 0.0031 | 0.0 ± 0.0 | 0.1782 ± 0.007 |
| Equalized odds difference | 0.0758 ± 0.0102 | 0.0697 ± 0.0025 | 0.0 ± 0.0 | 0.0706 ± 0.0069 | 0.0829 ± 0.0072 | 0.0 ± 0.0 | 0.0698 ± 0.0046 |
| Precision difference | 0.0181 ± 0.0123 | 0.0182 ± 0.0041 | N/A | 0.0162 ± 0.0117 | 0.0125 ± 0.0136 | N/A | 0.006 ± 0.0044 |
| Overall AUC-PR | 0.7922 ± 0.0016 | 0.7709 ± 0.0007 | 0.7667 ± 0.0003 | 0.7796 ± 0.0005 | 0.7727 ± 0.0014 | 0.772 ± 0.0008 | 0.7827 ± 0.0008 |
| AUC-PR difference | 0.0684 ± 0.009 | 0.069 ± 0.0016 | 0.0686 ± 0.0012 | 0.0697 ± 0.0017 | 0.0711 ± 0.0017 | 0.0727 ± 0.0016 | 0.0682 ± 0.002 |
| Acceptance rate difference | 0.1775 ± 0.0162 | 0.1786 ± 0.0046 | 0.1676 ± 0.0029 | 0.1682 ± 0.0044 | 0.1784 ± 0.0095 | 0.1811 ± 0.0049 | 0.1733 ± 0.0139 |
| Equalized odds difference | 0.076 ± 0.0197 | 0.0779 ± 0.0057 | 0.0714 ± 0.0031 | 0.0653 ± 0.0027 | 0.0799 ± 0.0135 | 0.0764 ± 0.0047 | 0.0688 ± 0.0097 |
| Precision difference | 0.017 ± 0.0073 | 0.0159 ± 0.0058 | 0.0243 ± 0.0084 | 0.0088 ± 0.0047 | 0.0103 ± 0.0051 | 0.0062 ± 0.0048 | 0.008 ± 0.0071 |

Table 11: Detailed results for the LSAC dataset. Best/worst C denotes the clipping norm with the respective highest/lowest performance.

| | Tuned SGD | Untuned DPSGD (best C) | Untuned DPSGD (worst C) | Tuned DPSGD | Untuned DPSGD-G.-A. (best C) | Untuned DPSGD-G.-A. (worst C) | Tuned DPSGD-G.-A. |
|---|---|---|---|---|---|---|---|
| Overall accuracy | 0.8582 ± 0.0007 | 0.8236 ± 0.0024 | 0.8152 ± 0.0023 | 0.8298 ± 0.0014 | 0.8255 ± 0.0009 | 0.8267 ± 0.0009 | 0.8297 ± 0.0009 |
| Accuracy difference | 0.0201 ± 0.0018 | 0.0215 ± 0.0011 | 0.0206 ± 0.001 | 0.092 ± 0.002 | 0.0205 ± 0.0013 | 0.0214 ± 0.0026 | 0.0183 ± 0.0028 |
| Acceptance rate difference | 0.0031 ± 0.0021 | 0.0022 ± 0.001 | 0.0019 ± 0.0011 | 0.005 ± 0.0009 | 0.0071 ± 0.0064 | 0.0084 ± 0.0054 | 0.0039 ± 0.002 |
| Equalized odds difference | 0.0413 ± 0.0101 | 0.0062 ± 0.0048 | 0.0043 ± 0.0027 | 0.0051 ± 0.0048 | 0.0341 ± 0.0153 | 0.0399 ± 0.0177 | 0.0196 ± 0.0101 |
| Precision difference | 0.0244 ± 0.0021 | 0.0228 ± 0.0008 | 0.022 ± 0.0008 | 0.0216 ± 0.0006 | 0.026 ± 0.0022 | 0.027 ± 0.003 | 0.0233 ± 0.0016 |
| Overall AUC-ROC | 0.8303 ± 0.0016 | 0.7185 ± 0.0012 | 0.7089 ± 0.0036 | 0.7449 ± 0.011 | 0.735 ± 0.0018 | 0.733 ± 0.0019 | 0.7504 ± 0.0026 |
| AUC-ROC difference | 0.0351 ± 0.0049 | 0.0388 ± 0.0016 | 0.0449 ± 0.0045 | 0.0377 ± 0.016 | 0.0323 ± 0.0028 | 0.0309 ± 0.0024 | 0.0214 ± 0.0026 |
| Acceptance rate difference | 0.0031 ± 0.0021 | 0.0022 ± 0.001 | 0.0019 ± 0.0011 | 0.0075 ± 0.0041 | 0.0071 ± 0.0064 | 0.0084 ± 0.0054 | 0.0065 ± 0.0042 |
| Equalized odds difference | 0.0413 ± 0.0101 | 0.0062 ± 0.0048 | 0.0043 ± 0.0027 | 0.0466 ± 0.0173 | 0.0341 ± 0.0153 | 0.0399 ± 0.0177 | 0.0327 ± 0.0136 |
| Precision difference | 0.0244 ± 0.0021 | 0.0228 ± 0.0008 | 0.022 ± 0.0008 | 0.0281 ± 0.0028 | 0.026 ± 0.0022 | 0.027 ± 0.003 | 0.0254 ± 0.0021 |
| Overall AUC-PR | 0.9343 ± 0.0012 | 0.8864 ± 0.0012 | 0.8834 ± 0.0027 | 0.9072 ± 0.0014 | 0.8978 ± 0.0013 | 0.8967 ± 0.001 | 0.9095 ± 0.0018 |
| AUC-PR difference | 0.0321 ± 0.0051 | 0.0383 ± 0.0014 | 0.04 ± 0.0021 | 0.013 ± 0.0036 | 0.0356 ± 0.0025 | 0.0334 ± 0.0027 | 0.0241 ± 0.0011 |
| Acceptance rate difference | 0.0031 ± 0.0021 | 0.0022 ± 0.001 | 0.0019 ± 0.0011 | 0.0188 ± 0.0185 | 0.0071 ± 0.0064 | 0.0084 ± 0.0054 | 0.0065 ± 0.0042 |
| Equalized odds difference | 0.0413 ± 0.0101 | 0.0062 ± 0.0048 | 0.0043 ± 0.0027 | 0.0471 ± 0.0362 | 0.0341 ± 0.0153 | 0.0399 ± 0.0177 | 0.0327 ± 0.0136 |
| Precision difference | 0.0244 ± 0.0021 | 0.0228 ± 0.0008 | 0.022 ± 0.0008 | 0.024 ± 0.0087 | 0.026 ± 0.0022 | 0.027 ± 0.003 | 0.0254 ± 0.0021 |

Table 12: Detailed results for the Compas dataset. Best/worst C denotes the clipping norm with the respective highest/lowest performance.

| | Tuned SGD | Untuned DPSGD (best C) | Untuned DPSGD (worst C) | Tuned DPSGD | Untuned DPSGD-G.-A. (best C) | Untuned DPSGD-G.-A. (worst C) | Tuned DPSGD-G.-A. |
|---|---|---|---|---|---|---|---|
| Overall accuracy | 0.6533 ± 0.0024 | 0.5462 ± 0.0183 | 0.5297 ± 0.009 | 0.6477 ± 0.0043 | 0.5594 ± 0.0223 | 0.543 ± 0.0096 | 0.6463 ± 0.0057 |
| Accuracy difference | 0.049 ± 0.0166 | 0.1152 ± 0.0174 | 0.1334 ± 0.0078 | 0.0266 ± 0.0144 | 0.0953 ± 0.0389 | 0.1192 ± 0.0147 | 0.0259 ± 0.0101 |
| Acceptance rate difference | 0.2636 ± 0.0108 | 0.0503 ± 0.0313 | 0.02 ± 0.0137 | 0.2703 ± 0.0198 | 0.1126 ± 0.07 | 0.048 ± 0.0195 | 0.2878 ± 0.0101 |
| Equalized odds difference | 0.2437 ± 0.0263 | 0.0725 ± 0.0412 | 0.0294 ± 0.0186 | 0.285 ± 0.0228 | 0.1278 ± 0.0753 | 0.0618 ± 0.028 | 0.3031 ± 0.0186 |
| Precision difference | 0.0358 ± 0.026 | N/A | N/A | 0.0759 ± 0.0319 | 0.0999 ± 0.0577 | 0.1558 ± 0.0542 | 0.0744 ± 0.0164 |
| Overall AUC-ROC | 0.6982 ± 0.0031 | 0.6894 ± 0.0023 | 0.5017 ± 0.0373 | 0.7035 ± 0.0025 | 0.6729 ± 0.0074 | 0.4935 ± 0.0349 | 0.6924 ± 0.0045 |
| AUC-ROC difference | 0.0081 ± 0.0051 | 0.0239 ± 0.0087 | 0.0331 ± 0.0189 | 0.0071 ± 0.0077 | 0.0178 ± 0.0174 | 0.0119 ± 0.0065 | 0.0116 ± 0.0082 |
| Acceptance rate difference | 0.2741 ± 0.0138 | 0.331 ± 0.0362 | 0.2287 ± 0.0987 | 0.2828 ± 0.0169 | 0.3074 ± 0.0481 | 0.1054 ± 0.0801 | 0.2791 ± 0.0147 |
| Equalized odds difference | 0.2880 ± 0.0172 | 0.3876 ± 0.046 | 0.2518 ± 0.1019 | 0.3052 ± 0.0261 | 0.3469 ± 0.0675 | 0.1185 ± 0.073 | 0.2839 ± 0.033 |
| Precision difference | 0.0785 ± 0.0223 | 0.0967 ± 0.0362 | 0.1325 ± 0.0399 | 0.0811 ± 0.0234 | 0.1201 ± 0.0726 | 0.1322 ± 0.0178 | 0.0691 ± 0.0405 |
| Overall AUC-PR | 0.6338 ± 0.0028 | 0.6603 ± 0.0099 | 0.5123 ± 0.0187 | 0.6919 ± 0.0014 | 0.6189 ± 0.0066 | 0.4925 ± 0.0247 | 0.6779 ± 0.0026 |
| AUC-PR difference | 0.1362 ± 0.0114 | 0.1656 ± 0.0154 | 0.1236 ± 0.0295 | 0.1453 ± 0.0067 | 0.183 ± 0.0224 | 0.1593 ± 0.0284 | 0.1248 ± 0.0161 |
| Acceptance rate difference | 0.31 ± 0.0099 | 0.2622 ± 0.0079 | 0.1345 ± 0.0838 | 0.3343 ± 0.0098 | 0.2849 ± 0.0742 | 0.1843 ± 0.1366 | 0.2821 ± 0.0178 |
| Equalized odds difference | 0.3456 ± 0.0193 | 0.2938 ± 0.0176 | 0.1517 ± 0.077 | 0.3716 ± 0.0117 | 0.3273 ± 0.0897 | 0.215 ± 0.1483 | 0.3033 ± 0.0379 |
| Precision difference | 0.0855 ± 0.0269 | 0.0991 ± 0.0395 | 0.1155 ± 0.0232 | 0.0621 ± 0.0191 | 0.1236 ± 0.04 | 0.1636 ± 0.0557 | 0.0882 ± 0.024 |

Table 13: Detailed results for the ACSEmployment dataset. Best/worst C denotes the clipping norm with the respective highest/lowest performance.

| | Tuned SGD | Untuned DPSGD (best C) | Untuned DPSGD (worst C) | Tuned DPSGD | Untuned DPSGD-G.-A. (best C) | Untuned DPSGD-G.-A. (worst C) | Tuned DPSGD-G.-A. |
|---|---|---|---|---|---|---|---|
| Overall accuracy | 0.8334 ± 0.0009 | 0.8319 ± 0.0003 | 0.6626 ± 0.008 | 0.8325 ± 0.0002 | 0.832 ± 0.0006 | 0.6901 ± 0.0124 | 0.8328 ± 0.0008 |
| Accuracy difference | 0.0076 ± 0.0047 | 0.0046 ± 0.0028 | 0.0882 ± 0.0548 | 0.0069 ± 0.0015 | 0.0026 ± 0.0017 | 0.0689 ± 0.0146 | 0.0042 ± 0.0034 |
| Acceptance rate difference | 0.3846 ± 0.0069 | 0.414 ± 0.0081 | 0.1693 ± 0.0813 | 0.4098 ± 0.0106 | 0.4217 ± 0.0099 | 0.2177 ± 0.1117 | 0.4144 ± 0.011 |
| Equalized odds difference | 0.4177 ± 0.0231 | 0.4766 ± 0.0273 | 0.1396 ± 0.0784 | 0.4763 ± 0.0395 | 0.5119 ± 0.0207 | 0.1825 ± 0.136 | 0.5014 ± 0.0207 |
| Precision difference | 0.183 ± 0.0201 | 0.1806 ± 0.0074 | 0.2887 ± 0.0677 | 0.1834 ± 0.0237 | 0.1819 ± 0.0158 | 0.3157 ± 0.0527 | 0.1895 ± 0.0263 |
| Overall AUC-ROC | 0.9124 ± 0.0004 | 0.9059 ± 0.0004 | 0.7519 ± 0.0109 | 0.9071 ± 0.0007 | 0.908 ± 0.0004 | 0.7647 ± 0.0142 | 0.9107 ± 0.0002 |
| AUC-ROC difference | 0.0347 ± 0.0025 | 0.0448 ± 0.0015 | 0.0432 ± 0.0071 | 0.0431 ± 0.0018 | 0.0432 ± 0.0033 | 0.0497 ± 0.0329 | 0.0414 ± 0.0058 |
| Acceptance rate difference | 0.3846 ± 0.0069 | 0.414 ± 0.0081 | 0.1693 ± 0.0813 | 0.4098 ± 0.0106 | 0.4217 ± 0.0099 | 0.2177 ± 0.1117 | 0.4184 ± 0.0121 |
| Equalized odds difference | 0.4177 ± 0.0231 | 0.4766 ± 0.0273 | 0.1396 ± 0.0784 | 0.4763 ± 0.0395 | 0.5119 ± 0.0207 | 0.1825 ± 0.136 | 0.4989 ± 0.0552 |
| Precision difference | 0.183 ± 0.0201 | 0.1806 ± 0.0074 | 0.2887 ± 0.0677 | 0.1834 ± 0.0237 | 0.1819 ± 0.0158 | 0.3157 ± 0.0527 | 0.1517 ± 0.0398 |
| Overall AUC-PR | 0.8858 ± 0.0006 | 0.8766 ± 0.0007 | 0.6932 ± 0.0194 | 0.878 ± 0.0009 | 0.8791 ± 0.001 | 0.6926 ± 0.0225 | 0.8835 ± 0.0002 |
| AUC-PR difference | 0.2884 ± 0.0075 | 0.3321 ± 0.0094 | 0.3135 ± 0.0067 | 0.3309 ± 0.0046 | 0.3276 ± 0.0106 | 0.3234 ± 0.0503 | 0.3204 ± 0.016 |
| Acceptance rate difference | 0.3846 ± 0.0069 | 0.414 ± 0.0081 | 0.1693 ± 0.0813 | 0.4098 ± 0.0106 | 0.4217 ± 0.0099 | 0.2177 ± 0.1117 | 0.4184 ± 0.0121 |
| Equalized odds difference | 0.4177 ± 0.0231 | 0.4766 ± 0.0273 | 0.1396 ± 0.0784 | 0.4763 ± 0.0395 | 0.5119 ± 0.0207 | 0.1825 ± 0.136 | 0.4989 ± 0.0552 |
| Precision difference | 0.183 ± 0.0201 | 0.1806 ± 0.0074 | 0.2887 ± 0.0677 | 0.1834 ± 0.0237 | 0.1819 ± 0.0158 | 0.3157 ± 0.0527 | 0.1517 ± 0.0398 |

Table 14: Detailed results for the MNIST dataset. Best/worst C denotes the clipping norm with the respective highest/lowest performance.

| | Tuned SGD | Untuned DPSGD (best C) | Untuned DPSGD (worst C) | Tuned DPSGD | Untuned DPSGD-G.-A. (best C) | Untuned DPSGD-G.-A. (worst C) | Tuned DPSGD-G.-A. |
|---|---|---|---|---|---|---|---|
| Overall accuracy | 0.9781 ± 0.0023 | 0.8976 ± 0.0012 | 0.8952 ± 0.0061 | 0.9187 ± 0.0032 | 0.9011 ± 0.0044 | 0.9007 ± 0.0077 | 0.9233 ± 0.0023 |
| Accuracy difference | 0.0883 ± 0.0144 | 0.3335 ± 0.0169 | 0.3439 ± 0.0631 | 0.3626 ± 0.047 | 0.3222 ± 0.024 | 0.3283 ± 0.0563 | 0.2557 ± 0.0123 |
| Precision difference | 0.0324 ± 0.0073 | 0.0549 ± 0.0173 | 0.0716 ± 0.022 | 0.0859 ± 0.0258 | 0.0741 ± 0.0186 | 0.0624 ± 0.0197 | 0.0596 ± 0.0111 |
| Overall AUC-ROC | 0.9996 ± 0.0001 | 0.9915 ± 0.0007 | 0.9909 ± 0.0004 | 0.9949 ± 0.0006 | 0.9925 ± 0.0007 | 0.9922 ± 0.0003 | 0.9953 ± 0.0005 |
| AUC-ROC difference | 0.0014 ± 0.0002 | 0.0115 ± 0.005 | 0.0097 ± 0.0032 | 0.0126 ± 0.0042 | 0.0105 ± 0.0017 | 0.0095 ± 0.0016 | 0.0032 ± 0.0015 |
| Precision difference | 0.0324 ± 0.0073 | 0.0563 ± 0.0144 | 0.0716 ± 0.022 | 0.0859 ± 0.0258 | 0.0624 ± 0.0197 | 0.0796 ± 0.0139 | 0.0551 ± 0.013 |
| Overall AUC-PR | 0.9977 ± 0.0003 | 0.9545 ± 0.0021 | 0.9521 ± 0.0014 | 0.9765 ± 0.0019 | 0.9549 ± 0.0038 | 0.9537 ± 0.0027 | 0.9767 ± 0.0018 |
| AUC-PR difference | 0.0044 ± 0.0011 | 0.052 ± 0.0122 | 0.0523 ± 0.0044 | 0.0215 ± 0.0056 | 0.0518 ± 0.0143 | 0.0555 ± 0.0135 | 0.0259 ± 0.0025 |
| Precision difference | 0.0386 ± 0.0196 | 0.077 ± 0.0197 | 0.0682 ± 0.0115 | 0.0518 ± 0.0089 | 0.0859 ± 0.0168 | 0.0679 ± 0.0189 | 0.0596 ± 0.0111 |

Table 15: Detailed results for the CelebA dataset. Best/worst C denotes the clipping norm with the respective highest/lowest performance.

| | Tuned SGD | Untuned DPSGD (best C) | Untuned DPSGD (worst C) | Tuned DPSGD | Untuned DPSGD-G.-A. (best C) | Untuned DPSGD-G.-A. (worst C) | Tuned DPSGD-G.-A. |
|---|---|---|---|---|---|---|---|
| Overall accuracy | 0.9608 ± 0.0017 | 0.9382 ± 0.0031 | 0.9344 ± 0.0016 | 0.9438 ± 0.001 | 0.9421 ± 0.0011 | 0.9402 ± 0.0035 | 0.9458 ± 0.0017 |
| Accuracy difference | 0.0541 ± 0.0072 | 0.091 ± 0.0035 | 0.1002 ± 0.0103 | 0.0932 ± 0.0037 | 0.0844 ± 0.0056 | 0.085 ± 0.0056 | 0.0804 ± 0.0071 |
| Acceptance rate difference | 0.3992 ± 0.0034 | 0.3908 ± 0.012 | 0.3718 ± 0.0131 | 0.3907 ± 0.0084 | 0.3905 ± 0.0043 | 0.3925 ± 0.0137 | 0.3942 ± 0.0064 |
| Equalized odds difference | 0.1714 ± 0.0193 | 0.269 ± 0.0283 | 0.2416 ± 0.0174 | 0.2543 ± 0.0197 | 0.2396 ± 0.0114 | 0.2552 ± 0.0355 | 0.235 ± 0.0215 |
| Precision difference | 0.0117 ± 0.0066 | 0.01 ± 0.0035 | 0.0107 ± 0.0043 | 0.0209 ± 0.0044 | 0.0133 ± 0.0079 | 0.0095 ± 0.0054 | 0.0147 ± 0.0058 |
| Overall AUC-ROC | 0.9929 ± 0.0005 | 0.9826 ± 0.0006 | 0.9791 ± 0.0013 | 0.9851 ± 0.0004 | 0.985 ± 0.0003 | 0.9847 ± 0.0008 | 0.9871 ± 0.0003 |
| AUC-ROC difference | 0.0333 ± 0.0052 | 0.0874 ± 0.003 | 0.1055 ± 0.0093 | 0.0734 ± 0.0026 | 0.0745 ± 0.0038 | 0.0779 ± 0.003 | 0.0645 ± 0.004 |
| Acceptance rate difference | 0.3992 ± 0.0034 | 0.3832 ± 0.0075 | 0.3718 ± 0.0131 | 0.3837 ± 0.0128 | 0.3905 ± 0.0043 | 0.3925 ± 0.0137 | 0.3942 ± 0.0064 |
| Equalized odds difference | 0.1714 ± 0.0193 | 0.2542 ± 0.0205 | 0.2416 ± 0.0174 | 0.2327 ± 0.0383 | 0.2396 ± 0.0114 | 0.2552 ± 0.0355 | 0.235 ± 0.0215 |
| Precision difference | 0.0117 ± 0.0066 | 0.0081 ± 0.0015 | 0.0107 ± 0.0043 | 0.0127 ± 0.0056 | 0.0133 ± 0.0079 | 0.0095 ± 0.0054 | 0.0147 ± 0.0058 |
| Overall AUC-PR | 0.9895 ± 0.0009 | 0.9731 ± 0.001 | 0.9682 ± 0.0017 | 0.9769 ± 0.0008 | 0.9762 ± 0.0007 | 0.9757 ± 0.0012 | 0.9795 ± 0.0005 |
| AUC-PR difference | 0.003 ± 0.0014 | 0.0118 ± 0.0015 | 0.0146 ± 0.0027 | 0.009 ± 0.0013 | 0.0097 ± 0.0017 | 0.0109 ± 0.0015 | 0.0082 ± 0.0022 |
| Acceptance rate difference | 0.3992 ± 0.0034 | 0.3832 ± 0.0075 | 0.3718 ± 0.0131 | 0.3837 ± 0.0128 | 0.3905 ± 0.0043 | 0.3925 ± 0.0137 | 0.3942 ± 0.0064 |
| Equalized odds difference | 0.1714 ± 0.0192 | 0.2542 ± 0.0205 | 0.2416 ± 0.0174 | 0.2327 ± 0.0383 | 0.2396 ± 0.0114 | 0.2552 ± 0.0355 | 0.235 ± 0.0215 |
| Precision difference | 0.0117 ± 0.0066 | 0.0081 ± 0.0015 | 0.0107 ± 0.0043 | 0.0127 ± 0.0056 | 0.0133 ± 0.0079 | 0.0095 ± 0.0054 | 0.0147 ± 0.0058 |

