# OpenReview forum: "Private and Fair Machine Learning: Revisiting the Disparate Impact of Differentially Private SGD"
_TMLR — Accepted by TMLR_

### Review · Reviewer_ziH6 · 2025-01-15

**Summary Of Contributions:**

I sincerely apologize for the delay in reviewing this work; I am still dealing with a family health issue.

----------------------------------------

This work evaluates the performance of SGD, DPSGD, and DPSGD-Global-Adapt (DPSGD-GA) across five datasets, various hyperparameter settings, and multiple metrics of fairness and classification performance. The authors investigate two scenarios: using the optimal hyperparameters selected for SGD with DPSGD versus optimizing DPSGD separately. The authors also include in the comparison the classification / fairness performance of DPSGD-GA.

The experiments demonstrate that:
	1.	DPSGD does not consistently outperform SGD across different fairness metrics.
	2.	For certain hyperparameters, DPSGD is competitive with SGD in terms of accuracy.
	3.	DPSGD-GA does not consistently surpass DPSGD in fairness.

**Audience:**

Yes

**Claims And Evidence:**

Yes

**Requested Changes:**

Not a change, but just a comment: A point is made in the text (after Figure 2.) about how the DPSGD line is smoother when sorted by its performance. Isn't the smoothness expected when you sort like that, even for a random set of points? It's possible I misunderstood the point there.

Suggested Changes

(These suggestions will not affect the decision, but may improve the presentation.)

1.	It would be helpful for the reader if the type of intervals used in the plots were mentioned. As I understand, there are five data points per element (one per 5-fold validation?), so are these intervals driven by standard deviation?
2.	In some tables, checkmarks indicate improvement, while in others they indicate a negative impact. Could the authors consider using, for instance, “x” for negative and a checkmark for improvement to enhance clarity?
3.	Please add a reference to DPSGD-Global-Adapt where it is first introduced in the paper’s introduction. While there is a reference later in the text, it would be more convenient for the reader to have it earlier.
4.	Is it feasible to briefly mention how DPSGD-Global-Adapt differs from standard DPSGD in the paper?
5.	Mentioning the thresholds used for "fairer / similarly fair / unfairer" (and similarly for accuracy) may be useful to readers.

**Strengths And Weaknesses:**

### Strengths

- I’m not versed enough in DP to comment on novelty, but I find the thesis of the paper clear enough: trying to shed some light on how metrics, methods, and data interrelate when it comes to fairness and classification performance.
- I appreciated that key takeaways are spelled out in the paper.
- The experiments are tracking a large variety of metrics; this would make this paper a useful reference for future experiments.

### Weaknesses

1. The paper would benefit from more discussion on the confidence of the results. The authors mention that they carried out Welch’s t-tests to check for significance. I assume this was carried over the results of the 5-fold cross-validation? While there is little variability between the methods when accuracy is close to its max value, the same is not true about accuracy difference. Would there be any benefit in adding more folds to the mix?

2. I feel that this paper had a chance to go a bit further and discuss the effects some of those hyperparameters have on fairness in a bit more detail, but maybe this goes beyond the goals the authors have for this work.

---

> ### Author Response · Authors · 2025-06-30
>
> Thank you for your careful and constructive feedback.
>
> 1.	As you correctly noted, the t-tests were carried out over the results of the 5-fold cross-validation. We decided to adopt the same number of folds as de Oliveira et al. (2023) to achieve a practical trade-off between statistical reliability and computational cost. A higher number of folds would significantly increase the already high computational cost and still does not necessarily result in less variability as smaller validation sets result in less reliable estimates, especially for small datasets or datasets where group imbalances are high. Moreover, in contrast to de Oliveira et al. (2023), we use significance tests rather than direct mean comparison to take the variability into account.
> For these reasons we do not expect our conclusions to change with a higher number of folds, but should there be remaining concerns, we would welcome a pointer to a particular result or comparison where you believe our conclusions might be unreliable due to high variability of the accuracy difference so we can examine this further.
>
> 2.	We appreciate the reviewers' suggestion regarding the impact of individual hyperparameters on fairness. While this is an interesting area, our work specifically investigates the implications of **hyperparameter tuning** on fairness in machine learning. We believe this focus is justified as isolating the impact of single hyperparameters is challenging, not only in private ML settings but in general, and some initial insights into this topic have been provided by Bagdasaryan et al. (2019).
>
> Regarding your comment on the discussion of Figures 1-5: You are right, the smoothness of the dashed orange line (DPSGD ordered by DPSGD accuracy) is expected. What is interesting, however, is that the dash-dot green line (DPSGD ordered by SGD accuracy) exhibits significant irregularity (with exception of the LSAC dataset, as stated). We re-formulated the sentence in question to clarify that the surprising result is not that DPSGD ordered by DPSGD accuracy is smoother than DPSGD ordered by SGD accuracy, but rather the magnitude of fluctuations of the latter.
>
> Thank you also for your additional suggestions, we agree that they will improve understandability of our work and incorporated them in the revised version of the paper (which will be uploaded as soon as we have received all reviews).

---

> > ### Comment · Reviewer_ziH6 · 2025-07-03
> > **Reply to authors**
> >
> > Thank you for the reply!
> >
> > It's been a while since I wrote this review, but I'll be waiting to take a look at the revised version!

---

### Review · Reviewer_mQAm · 2025-03-31

**Summary Of Contributions:**

This paper considers the interaction of DP-SGD and group fairness. Prior work has suggested that DP-SGD may result in less fair models compared to (non-private) SGD. Some studies have investigated how hyperparameter tuning affects the story: since DP-SGD may require, for example, a different learning rate than SGD, it may be more appropriate to tune the hyperparameters separately before comparing fairness metrics.

This submission continues that line of work and investigates hyperparameter tuning, group fairness, and privacy. They consider five datasets and six notions of group fairness. The two main experimental setups are: i) select the best hyperparameters non-privately, then run DP-SGD with the same hyperparameters, and ii) tune the two algorithms independently. There are also some experiments on "DPSGD-Global-Adapt," a recently proposed variation of DP-SGD intended to improve fairness.

**Audience:**

No

**Broader Impact Concerns:**

None.

**Claims And Evidence:**

No

**Requested Changes:**

Before recommending this paper for acceptance, I would like a revision that helps me understand:
1. What new insights are in this paper,
1. Who these insights are helpful for, and
1. How they differ from existing knowledge.

It may be the case that, with substantial effort to clarify the results and takeaways, this paper will be a solid contribution. As it stands, I have a hard time arguing that any substantial part of the TMLR audience will find it useful.

**Strengths And Weaknesses:**

I think this paper needs substantial revisions before it is accepted. There are many low-level experimental observations that fail to connect to the larger story. Because the results depend so heavily on the dataset, learning algorithm, and notion of fairness, I feel the paper fails to identify trends or deliver insights that would be useful for someone wishing to train a fair and private neural network. It is possible that the experiments contain these insights, but they didn't not come through to me in the discussion.

Setting practitioners aside, it is not clear to me what this work contributes to our understanding of privacy and fairness. The submission fails to make a convincing argument that it adds to scientific knowledge: the overarching theme seems to be "it depends." I struggle to even call this a negative result, since I don't see a clear hypothesis being tested.

This submission largely extends experiments from prior work, and at times I was unclear about which results are actually new. For example, from the conclusion: "Our findings reveal that disparate impacts on different metrics do not necessarily co-occur, and that the impact of DPSGD is not consistent across a wide range of hyperparameter settings." As I understood it, both of these statements are known from prior work. We know that some metrics are incompatible and that the clipping norm plays a significant role in fairness.

It's possible I misunderstand the submission. I look forward to reading the other reviews.

Minor comments
- I found the line plots in Figues 1-5 difficult to parse. Once I did understand what was going on, I had trouble interpreting them.
- I suggest including a concise takeaway in each figure caption, or one takeaway per subfigure.
- In your significance tests, do you adjust for multiple comparisons? You should mention this, either way.
- "The Algorithmic Foundations of Differential Privacy" is just by Dwork and Roth, there's no "et al."
- Footnotes marks should go after the period to end a sentence. The submission is inconsistent here.

---

> ### Author Response · Authors · 2025-06-30
>
> Thank you for your review, we revised the manuscript to include the requested changes and improve clarity on our contributions (the revised version will be uploaded as soon as we have received all reviews).
>
> 1. New insights:
> We show that the claim that performance-based hyperparameter tuning can mitigate the disparate impact of DPSGD is not generalizable. Nevertheless, our findings lead to the conclusion that performance-based hyperparameter tuning remains a good practice, not only when training accurate but also accurate and fair models. Our results suggest that neither multi-objective hyperparameter tuning nor DPSGD-Global-Adapt (with or without hyperparameter tuning) can mitigate the disparate impact of DPSGD on accuracy.
>
> 2. Audience:
> We believe these insights offer an important contribution by highlighting the often-overlooked role of hyperparameter tuning when developing or deploying privacy-preserving models under fairness constraints. Our results underscore the need for researchers to systematically tune hyperparameters when proposing new methods and serve as a caution to practitioners against overreliance on proposed solutions whose dependence on data and hyperparameter choice is underexplored.
>
> 3. Difference from existing knowledge:
> While it is true that it has been shown before that some fairness metrics are incompatible, this is not what we are focusing on. We are focusing on how the disparate impact manifests across metrics. (i.e., instead of "unfairness in one metric does not imply unfairness in another" we look at "disparate impact on one metric does not imply disparate impact on another", a related but distinct question). As far as we know, we are the first to show this explicitly across a wide range of metrics. While de Oliveira et al. (2023) also use a similar variety of fairness metrics and shortly mention that hyperparameter tuning mitigates the disparate impact of DPSGD across opposing fairness metrics, they 1) do not discuss the co-occurrence to begin with, and 2) only use AUC-ROC and AUC-ROC equality, treating it as equivalent to accuracy and accuracy equality used in previous works.
> It is also true that previous work has shown that clipping norm plays a significant role in fairness and that the impact of DPSGD **on performance** is not consistent across hyperparameter settings. However, our paper shows that this generalizes to the impact of DPSGD **on fairness**.
> Moreover, our paper does not focus on DP-specific hyperparameters (such as clipping norm) but on general hyperparameters of neural networks - in fact, we aimed to remove the impact of clipping norm as much as possible by consistently reporting the results with highest performing clipping norm (and for comparison the lowest performing clipping norm in the appendix).
>
> We also addressed the minor comments as follows:
>
> •	We added takeaways to the figure captions to improve interpretability of the plots.
>
> •	We clarified that we do not adjust for multiple comparisons as we consider false positives and false negatives to be equally undesirable and want to avoid overly conservative adjustments that could distort the balance between the two.
>
> •	We fixed the reference typo and footnote inconsistency.
>
> Thank you again for your time and effort.

---

### Review · Reviewer_1dwj · 2025-07-02

**Summary Of Contributions:**

The paper investigates how both Differentially Private Stochastic Gradient Descent (DPSGD) and DPSGD-Global-Adapt affect performance and fairness metrics for neural networks. To do so, it empirically analyzes neural networks across different datasets (3 tabular, 2 image) and hyperparameters (e.g., clipping norm, learning rate) using a range of metrics. One of the main con that DPSGD’s disparate impacts on fairness (e.g., accuracy disparity vs. AUC disparity) do not consistently co-occur which challenges prior results. Likewise, the paper shows that although not consistent, hyperparameter tuning can bring more reliable results when compared to simply reusing the same parameters as the non-DP model. Finally, it shows that DPSGD-Global-Adapt does not bring significant advantages over DPSGD in terms of fairness.

**Audience:**

Yes

**Claims And Evidence:**

Yes

**Requested Changes:**

See weaknesses section.

**Strengths And Weaknesses:**

Strengths:
- Paper is well structured and reasoning is easy to follow - particular good point given to the 3 research questions outlined in the beginning which are each answered and discussed
- Explanations are clear and analyses are rigorous - good nuance in the different performance metrics used and comprehensiveness of the parameter set explored

Weaknesses:
- The core findings are basically "it depends on the dataset/metric/hyperparameters", which lacks some depth. The paper does catalogs inconsistencies but doesn't give any explanation or mechanism or theoretical exploration behind these inconsistencies.
- Only 5 datasets are tested, with toy examples examples (MNIST) and small tabular datasets (Adult, Compas). For a paper that is primarily empirical and aims at benchmarking, this is not enough - and not enough real-world complexity (imbalance, large scale etc.)
- The recommendations of "tuning hyperparameters help" are obvious and already adopted in practice. I would have wanted more light on the why/how one should tune their hyperparameters to balance fairness and privacy.

---

> ### Author Response · Authors · 2025-07-16
>
> Thank you for your review.
> - We acknowledge that our work does not provide a theoretical explanation. However, we believe that understanding the empirical inconsistencies is a necessary first step toward any deeper theoretical modeling. Moreover, hyperparameter tuning is, in general, a highly empirical process. In practical settings it is often guided by exploratory search over combinations, rather than systematic isolation of individual hyperparameter effects. Thus, we believe that explicitly demonstrating that performance-based tuning can improve the fairness of DPSGD, while also highlighting its limitations, offers a valuable contribution.
> - We chose standard, well-established benchmarks from a variety of domains to ensure our results are comparable to prior work. That said, we appreciate the concern about dataset scale and complexity. To address this, we’ve extended our evaluation to include an additional dataset, ACSEmployment, which is over 2.5 times larger than the previously considered tabular datasets, and evaluated it on the highly imbalanced protected attribute "vision difficulty", to better reflect real-world challenges.
> - We understand that our conclusion that hyperparameter tuning improves outcomes might seem obvious. However, we respectfully disagree that it is widely adopted in practice – at least in research settings - as we showcased with DPSGD-Global-Adapt. Our experiments demonstrate that simple tuning strategies, such as performance-based grid or random search, can effectively reduce accuracy disparities. At the same time, we find that the extent to which hyperparameter tuning can reduce these disparities is limited, emphasizing the need for novel strategies to train private and fair ML models.

---

### Author Response · Authors · 2025-07-16

We thank the editor and all three reviewers for their valuable comments and suggestions. We have incorporated them into the revised manuscript (changes are marked in red).

---

### Decision · Action_Editor_pLt4 · 2025-09-01

**Recommendation:** Accept with minor revision

**Audience:**

Yes

**Audience Explanation:**

The impact of DP-SGD on fairness is of particular interest to a subset of TMLR’s readership, and I believe the experiments presented in this paper would be highly relevant to that community.

**Claims And Evidence:**

Yes

**Claims Explanation:**

The paper presents an experimental study evaluating the validity of prior work’s hypothesis regarding the effect of DP-SGD on fairness. The reviews are borderline: while some reviewers request additional experiments, I believe that with more precise interpretation and careful presentation, the existing results sufficiently support the paper’s claims. I recommend acceptance, conditional on the following minor revisions:

- The paper should articulate its claims with greater precision and restraint. Overgeneralizations should be avoided. For example, in the title, I recommend explicitly referring to DP-SGD (or another descriptive term) rather than the broader word privacy. There are multiple approaches to achieving private models, and DP-SGD represents only one of them.

- Given that this is an experimental study, the conclusions should be phrased more cautiously. For instance, in the Takeaways boxes or other parts of the paper, instead of categorical statements like “does not,” it would be better to use phrasing such as “may not,” reflecting that the observations are limited to the current set of experiments.

- The discussion of hyperparameter tuning should be revised. In DP-SGD, hyperparameter tuning can weaken differential privacy guarantees, since each additional trial increases the effective values of $\epsilon$ and $\delta$. The paper should therefore frame this issue with caution and avoid recommending hyperparameter tuning for DP-SGD. To prevent misinterpretation, this point should also be emphasized in both the abstract and conclusion, making it clear that hyperparameter tuning is not being advocated.